# Theta-gamma phase amplitude coupling serves as a marker of social cognition and visual working memory deficits in individuals with elevated autistic traits

Elisabeth V. C. Friedrich [1,2,3] ✉, Yannik Hilla [3,4] ✉, Elisabeth F. Sterner[2,5], Simon S. Ostermeier[2], Larissa Behnke [3,6] & Paul Sauseng[3,6]

It has been thought that coordination of briefly maintained information (working memory) and social cognition (mentalizing) rely on different brain mechanisms. However, the dorsomedial prefrontal cortex (DMPFC) seems to control the mentalizing and the visual working memory networks. We aimed to show (1) that visual working memory and social cognition share the same neural communication mechanism (i.e., interregional phase-amplitude coupling) and (2) that this mechanism is behaviorally relevant. We analyzed electrical brain activity from 98 volunteers who differed in the extent of (subclinical) autistic personality traits. Participants performed a social, visual and verbal working memory task, each implemented in a low and a high cognitive load version. We analyzed how slow rhythmical brain activity in the DMPFC controls distributed posterior regions associated with working memory and mentalizing via phase-amplitude coupling. First, individuals with low autistic personality traits use slow rhythmical brain activity in the DMPFC to precisely tune communication with posterior brain areas depending on the effort necessary in the visual and social tasks. Second, individuals with high autistic personality traits struggle in fine-tuning this mechanism, which is associated with difficulties in efficiently adapting brain activity to the difficulty level of a visual working memory task; and they demonstrate problems with efficiently synchronizing the relevant cortical network in a social cognition task. While these findings suggest a unified function of brain oscillations in cognitive coordination between social and visual tasks, they could also explain why individuals with high autistic personality traits can have difficulties with demanding cognitive processing and mentalizing.

Imagine you are in charge of assigning the best-suited welcome present to each of your friends at a party. For this, you will need to infer from your friends' personality traits which type of present they might like; you will have to hold this information in memory and mentally combine friend-present pairings. Depending on how well you do these complex cognitive operations, the presents will either be a success or disappointment.

These operations - known as social cognition - involve storing social information and applying it to oneself and others. While subtle miscalculations can happen to everyone and may cause everyday uncomfortable moments, more persistent or severe difficulties in social cognition can have significant consequences and can be observed in various psychiatric disorders[1]. Thus, the investigation of the underlying neural mechanisms of social cognition is a crucial and clinically highly relevant research topic.

Attributing mental traits to another person, such as what kind of presents someone might like or dislike, relies on activity within the so-called

[1]Faculty of Psychology, University of Sustainability Vienna – Charlotte Fresenius Privatuniversität, Vienna, Austria. [2]Department of Psychology, Ludwig-Maximilians-Universität München, Munich, Germany. [3]Department of Psychology, Neuropsychology and Cognitive Neuroscience Unit, University of Zurich, Zurich, Switzerland. [4]Department of Human Sciences, Institute of Psychology, University of the Bundeswehr Munich, Neubiberg, Germany. [5]Department of Diagnostic and Interventional Neuroradiology, School of Medicine, Technical University of Munich, Munich, Germany. [6]Neuroscience Center Zurich, University of Zurich, Zurich, Switzerland. ✉e-mail: elisabeth.friedrich-higgs@uni-sustainability.at; yannik.hilla@psychologie.uzh.ch

mentalizing network in the brain[2–6]. It includes a group of cortical regions strongly associated with social cognition in the dorsal parietal cortex, temporal cortex, cingulate cortex, and dorsal and medial prefrontal cortex[2–6]. However, high-level social cognitive functioning usually also requires flexible transient storage and utilization of large quantities of information—processing known as working memory[7]. Accordingly, in situations as described in the example above, one would also expect an involvement of the working memory system in addition to the mentalizing network. It is, however, unclear how far the mentalizing and the working memory networks can be active in the brain in parallel: While there is research suggesting that they show an antagonistic relationship[8,9], Meyer and colleagues proposed that truely effortful social cognition relies on simultaneous activity of both fronto-temporoparietal mentalizing and working memory networks[10–12].

The dorsomedial prefrontal cortex (DMPFC) has been described as a hub region for social cognition[10] as well as a key region for (non-social) working memory processing[13–16]. This makes the DMPFC a prime candidate for linking the mentalizing network and the working memory system. And it is exactly the DMPFC which – under high cognitive demand – expresses slow (4 to 7 Hz), high amplitude rhythmical activity that can be measured using the human electroencephalogram (EEG). This so-called frontal-midline theta activity (FM-theta)[13–16] has been associated with cognitively demanding working memory tasks and central executive functions[17–20]. Moreover, FM-theta activity seems to play a key role in organizing and synchronizing brain activity across the cortex[17,21] and, thus, in establishing effective neural communication[22].

Coordination of neural communication between distributed cortical regions during demanding visual working memory processes was recently shown to be implemented via interregional phase-amplitude coupling[23,24]: In this way, fast frequency posterior cortical activity (in the gamma frequency range - which can be considered a proxy for working memory activity[25–27]) is nested into specific phases of the FM-theta cycle[23]. Berger and colleagues showed that this phase-amplitude coupling mechanism was working memory load-dependent[23]: In a visual working memory task requiring a lot of cognitive control, posterior gamma activity was nested into FM-theta trough[23]– the excitatory phase[28–30]. In contrast, in an easy visual working memory task, posterior gamma was nested more towards the rather inhibitory peak phase. This way, depending on how effortful a working memory process might be and dependent on how much cognitive control is necessary, fronto-parietal cortical networks can either be coupled (gamma amplitude peak in theta trough) or actively dis-engaged (gamma amplitude peak shifted away from theta trough) to preserve cognitive resources. Applying this oscillatory coupling/decoupling mechanism, the DMPFC, therefore, can be considered perfectly adjusted for interfacing the mentalizing and the visual working memory networks[10,23].

Controlling a distributed verbal working memory network[31] seems to be slightly different than coordinating mentalizing and visual working memory networks. There, the left dorsolateral prefrontal cortex (DLPFC) seems more relevant than the DMPFC[10,32–35]. In brain stimulation studies, the inhibition of the left DLPFC led to a performance decrease in a broad variety of tasks, suggesting a more general working memory function[34–36]. In contrast, an artificial lesion over the DMPFC was found to impact performance in visual working memory as well as mentalizing tasks[36,37].

Consequently, verbal working memory seems to recruit a distributed cortical network with upmost importance of the DLPFC. The visual working memory and mentalizing tasks seem to be coordinated by the DMPFC and to engage networks responsible for detecting, attending and processing visual stimuli, which also act as social cues (i.e., detecting gaze directions, reading facial expressions, reorienting visual attention)[38,39].

Coordinating mentalizing networks and working memory systems can be of particular importance in conditions with difficulty in social processing as well as transiently retaining and utilizing non-social information. This could be the case in autism (we based our choice of language on ref. 40). For autistic individuals, mentalizing and reasoning about others can be particularly challenging[41–43]. In addition, difficulties in non-social working memory tasks[44], especially in the visuo-spatial domain[45–51] have been observed in autistic individuals. Remarkably, even in the subclinical, general population the extent of autistic traits predicted visual working memory performance[46].

On the neural level, several studies found that autistic individuals displayed lower activity in the prefrontal cortex and long-range dysconnectivity, especially a fronto-posterior under-connectivity in lower frequencies[52–54]. Moreover, theta activity was found to be correlated with autistic characteristics in children without a diagnosis of autism[55].

While several studies indicate differences in brain activity in autistic individuals, there is a lack of evidence for behavioral effects in the typically used n-back tasks for measuring working memory performance[47,56,57]. These neural differences in autistic individuals could partly reflect efficient compensation mechanisms so that a similar level of behavioral performance as in nonautistic individuals can be achieved[58–60]. However, compensating mechanisms can reach their limit and difficulties in adjusting to cognitively more effortful tasks can become evident[45,58,60–62].

Then again, it is exactly this optimal allocation of neural resources depending on task-difficulty (in visual working memory processes) that is reflected by fine-tuned FM-theta phase to posterior gamma amplitude coupling, as discussed earlier[23]. With FM-theta being generated in the DMPFC that supposedly represents a hub region for coordinating mentalizing and working memory networks. There is also reason to believe that the phase-amplitude coupling mechanism responsible for coordinating communication between brain regions in visual working memory tasks can be extended to social cognition. This will shed light on the contradictory ideas how working memory and mentalizing are associated with each other[8,9,11,63,64]. Moreover, if a common neural communication pathway is the underlying cause for differences in social cognition and visual working memory in autism, this should also be reflected by FM-theta phase to posterior gamma amplitude coupling being associated with the extend of autistic traits in individuals.

This raises two research questions: (i) if effortful social cognition requires cooperation between a fronto-posterior mentalizing network and a fronto-posterior working memory network, would those networks be coordinated by the above described FM-theta phase mechanism? And if so, (ii) could this phase-amplitude coupling be behaviorally relevant and explain why some people have difficulties in social cognition as well as visual working memory?

Addressing research question (i), we expected that in a social and visual working memory task, fronto-temporoparietal networks would be coordinated by FM-theta phase from the DMPFC (in contrast to a verbal working memory task). We hypothesised that the mechanism of posterior brain activity nesting into different FM-theta phase segments depending on cognitive effort (see ref. 23) should be observable for the mentalizing network in the same way as for the working memory system in individuals with low autistic personality traits.

Regarding research question (ii), we expected participants with high autistic traits to perform worse in a social working memory task based on a failure of efficient nesting of posterior high frequency brain activity into the excitatory FM-theta phase. We also hypothesised that participants with high autistic traits would be more rigid in their dynamic control of fronto-temporoparietal networks in the visual working memory task and would show lower performance especially at the high-demand level of the task.

Data in line with these hypotheses would (i) provide evidence for a common mechanism of controlling visual working memory as well as mentalizing by means of FM-theta oscillations; and (ii) should explain why social cognition as well as visual working memory processes may differ depending on the extent of autistic traits.

## Methods
### Sample size
We based our sample size on an *a priori* power analysis (MorePower 6.0.4[65]) for a task (visual vs. social vs. verbal) x cognitive load (low vs. high) repeated-measures ANOVA with autistic-traits group as between-subject factor using

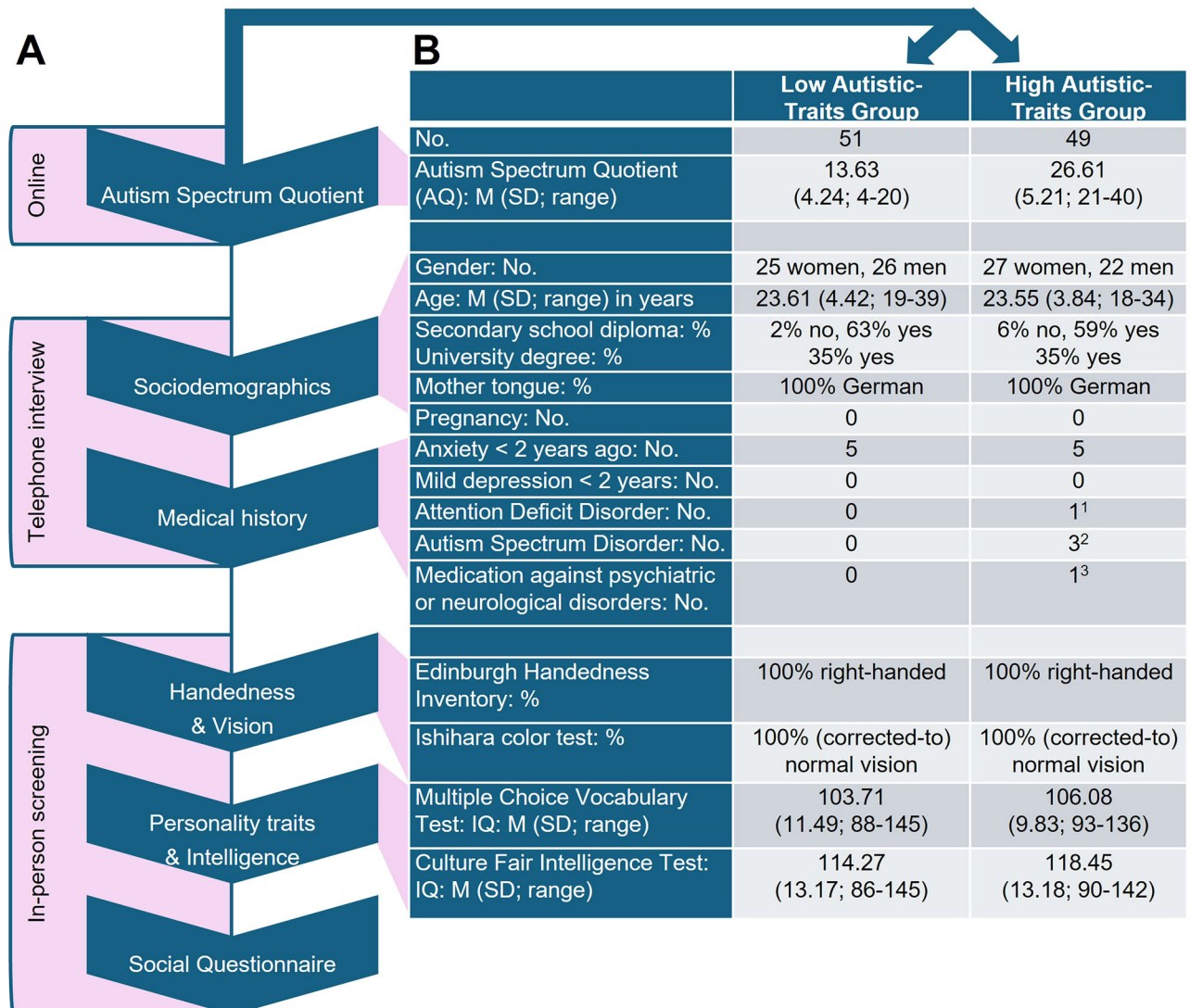

| | Low Autistic-Traits Group | High Autistic-Traits Group |
|---|---|---|
| No. | 51 | 49 |
| Autism Spectrum Quotient (AQ): M (SD; range) | 13.63 (4.24; 4-20) | 26.61 (5.21; 21-40) |
| | | |
| Gender: No. | 25 women, 26 men | 27 women, 22 men |
| Age: M (SD; range) in years | 23.61 (4.42; 19-39) | 23.55 (3.84; 18-34) |
| Secondary school diploma: % University degree: % | 2% no, 63% yes 35% yes | 6% no, 59% yes 35% yes |
| Mother tongue: % | 100% German | 100% German |
| Pregnancy: No. | 0 | 0 |
| Anxiety < 2 years ago: No. | 5 | 5 |
| Mild depression < 2 years: No. | 0 | 0 |
| Attention Deficit Disorder: No. | 0 | 1[1] |
| Autism Spectrum Disorder: No. | 0 | 3[2] |
| Medication against psychiatric or neurological disorders: No. | 0 | 1[3] |
| | | |
| Edinburgh Handedness Inventory: % | 100% right-handed | 100% right-handed |
| Ishihara color test: % | 100% (corrected-to) normal vision | 100% (corrected-to) normal vision |
| Multiple Choice Vocabulary Test: IQ: M (SD; range) | 103.71 (11.49; 88-145) | 106.08 (9.83; 93-136) |
| Culture Fair Intelligence Test: IQ: M (SD; range) | 114.27 (13.17; 86-145) | 118.45 (13.18; 90-142) |

**Fig. 1 | Screening and participants' characteristics. A** The screening process is indicated. **B** Number of participants (no.) and percentage of participants (%), mean (M), standard deviation (SD) and the range of the variables from the screening are indicated. [1]One person reported an Attention-Deficit/Hyperactivity Disorder without medication. [2]Three participants reported an Autism Spectrum Disorder in the high autistic-traits group. However, their diagnosis did not lead to clinically significant impairment in daily-life functioning. [3]No one took medication against psychiatric or neurological disorders, except for one person who used a low dosage of Escitalopram (7.5 mg).

an alpha level of 0.05, statistical power of 0.9 and an effect size of partial $\eta^2 = 0.07$. This effect size was a conservative estimation based on both relevant behavioral (effect size of $\eta^2 \geq 0.08$ for behavioral performance in the social and verbal tasks[10]) as well as EEG findings (partial $\eta^2 = 0.1$ for phase-amplitude coupling in the visual task[23]). We added 12 participants to the suggested minimum of 88 participants to compensate for possible exclusions, e.g., insufficient data quality, and thus, recorded data of a total of 100 participants.

## Autistic personality traits

Autism is a spectrum and includes a wide range of characteristics[66]. In some cases, these characteristics are clinically significant and lead to a formal diagnosis. But autistic traits can also be present in individuals who do not meet the full diagnostic criteria (i.e., subclinical range). As shown by a meta-analysis by ref. 67, the differences between individuals with and without a diagnosis have decreased over time, highlighting the importance of using subclinical trait measures to capture the whole spectrum. For this reason and also to control for possible confounding factors such as medication intake or co-occurring conditions, we recruited a subclinical sample. As the tasks used in this study were cognitively highly demanding and required a certain level

of social abilities, we recruited young adults and subdivided them into a low and high autistic-traits group based on their Autism Spectrum Quotient (see below for further information)[68]. Individuals of both groups were comparable in age, gender, education, intellect and medical history but differed in the extent of autistic personality traits (i.e., low versus high autistic-traits group, Fig. 1). To investigate whether the extent of autistic personality traits makes a difference in this otherwise very homogenous subclinical sample, the sample was dichotomized. By including the autistic-traits groups as between-subject factor in the regression model, we could test whether the difference in the extent of autistic-traits was a significant predictor variable for the load-dependent phase-amplitude coupling.

## Screening

Participants were mainly recruited within the Ludwig-Maximilians-Universität (LMU) Munich and Technical University Munich student communities and were compensated with course credit or 10 € per hour. All participants gave written informed consent prior to participation in the study, which was approved by the LMU Munich Faculty 11 ethics committee.

In our screening, first we used the Autism Spectrum Quotient (AQ)[68] (implemented on SoSci Survey), which assesses social skills, attention

switching, attention to detail, communication and imagination in five subscales. The scale is sensitive enough to measure autistic traits in the nonclinical population, and it correlates with (social) cognitive performance and neurophysiological measures[46,69–71]. We used the AQ to discriminate between participants with low autistic traits (i.e., AQ ≤ 20 for the low autistic-traits group) and participants showing high but subclinical autistic traits (i.e., AQ > 20 for the high autistic-traits group). We pre-defined the cut-off of 20 based on literature as *a priori* criterion[68,72,73] and selectively screened and selected participants so that half the sample consisted of participants AQ ≤ 20 and the other half of individuals with AQ > 20. Participants were blinded to their group allocation, whereas the investigators who were also responsible for recruitment and screening were not.

After assigning the participants to the low or to the high autistic-traits group according to their AQ[68], all participants were asked about their sociodemographic data and underwent a neuropsychiatric telephone interview to only include participants who had no clinically significant neurological or psychiatric disorder. The interview followed a structured set of questions focused on symptoms associated with psychiatric disorders. The exception was a diagnosis of Autism Spectrum Disorder for participants in the high autistic-traits group as long as participants were comparable to the other participants in age, education and intellect. Gender was determined based on information provided by the participants. No data on race or ethnicity was collected.

The last step was an in-person screening which took place on average 14 days prior to the EEG recording (see Fig. 1). It included testing handedness with the Edinburgh Handedness Inventory (short form[74]), color vision with the Ishihara color test, verbal intelligence with the Multiple Choice Vocabulary Test (MWT-B[75]), visuospatial problem solving with the Culture Fair Intelligence Test (CFT 20-R[76]), and the Big Five personality traits with the German short version of the Big Five Inventory (BFI-K[77]).

Finally, participants filled out the social questionnaire[10,12,78] (implemented on SoSci Survey). They named 10 personal contacts and evaluated these contact persons on 48 traits (e.g., sensitive, dominant[79]) on a 1–100 rating scale in intervals of 5 points. Personal contacts which were ranked ≥15 points apart from one another for a specific trait were used to create individualized trials for the social working memory task in the EEG experiment.

## Participants
Of the 100 volunteers, 51 belonged to the low autistic-traits group (AQ ≤ 20) with a mean (M) AQ of 13.63 and a standard deviation (SD) of 4.24. Typical control groups were reported to show an average AQ of 16.5 with SD up to 7.5[46,68,72,73,80,81].

The remaining 49 participants belonged to the high autistic-traits group and had an average AQ of 26.61 (SD = 5.21). Scores above 26 indicate that individuals have clinically significant levels of autistic traits[82]. In our high autistic-traits group, 20 participants scored above 26 and three participants reported a diagnosis of Autism Spectrum Disorder.

The two groups were comparable in their sociodemographics, medical history, handedness, vision, verbal intelligence (Multiple Choice Vocabulary Test; MWT-B; $t(98) = -1.11$, $p = 0.270$, two-tailed, CI = [−6.63, 1.88])[75] and visuospatial problem solving (Fair Intelligence Test; CFT 20-R; $t(98) = -1.58$, $p = 0.116$, two-tailed, CI = [−9.41, 1.06])[76] (see Fig. 1).

## Experimental design
The experimental design included a social, visual and verbal working memory task, each implemented in a low (i.e., two stimuli) and a high load condition (i.e., four stimuli to be encoded and manipulated) (Fig. 2). The social and verbal tasks were adapted from Meyer and colleagues[10,12].

In the social task, the names of the personal contacts - which were indicated in the social questionnaire during the screening process - were used as stimuli. After encoding the names for 4 s, a trait was presented for 1.5 s (e.g., shy), which was also drawn from the social questionnaire. During the 5-s manipulation period, participants mentally ranked their contacts according to the trait (i.e., how much the trait applies to the person). Names and traits varied between trials. The probe then asked if a certain name was

at a certain position in the mental ranking (e.g., Is Sue second shiest?). For example, the participant should press "no" if Sue was the shiest person (see Fig. 2). After the response or maximally 4 s, a 1-s break occurred, followed by a jittered inter-trial interval of 2 ± 0.5 s before the next trial started.

In the visual task, the stimuli consisted of 16 pictures of different fruits and vegetables, which naturally vary in color and size. Note that the stimuli were always presented in the same size, which was ensured by using the same amount of colored pixels to depict the objects against the grey background. The instruction was to sort the objects either with respect to color (from green to blue) or with respect to their natural size (from small to large). Both manipulation conditions appeared randomized and equally often. To ensure that participants rely on the same ranking of color and size, a uniform ranking was included in the preparation material, and was rehearsed before the beginning of the EEG experiment.

In the verbal task, names were used as stimuli. The names were drawn randomly from a pool of 16 German names, which all differed in their first and last letter. Participants should either sort the names in alphabetical order with respect to their first or last letter. Both manipulation conditions appeared randomized and equally often.

Each task and load condition was presented in a separate block of 16 task-specific trials. We also added 4 so-called order trials as control trials to each block to ensure that participants encoded all stimuli in the presented order and did not start to rank the stimuli before the manipulation period started. In the order trials, the encoding stimuli were the same as for the task-specific trials. However, instead of the task-specific instruction, participants were asked to remember the stimuli in the presented order without any manipulation. After a 1-s retention period, the probe then referred to the original position of the stimuli during the encoding period. Importantly, the order trials were not included in any behavioral or EEG data analyses.

Each block was presented twice in randomized order during the whole EEG session, which (depending on the reaction time) lasted for 66.8 min (without breaks) maximum.

Thus, we recorded in total 240 trials, i.e., 32 task-specific trials per task and load condition. As our time window to analyze phase-amplitude coupling was 2.5 s long per trial, this number of trials leaves us with over 13,000 data points per analyzed phase segment, ensuring robust analyses.

## Procedure and EEG recordings
After a successful screening, participants were invited to the EEG session. They were tested on the ranking of tasks (e.g., from smallest to biggest object). The ranking of the tasks and a description of the traits was sent to the participants as preparation material. They were provided with instructions and completed nine exercise trials for each of the tasks and load conditions.

Participants performed the experimental task on the computer (Presentation ® software, version 20.1, build 12.04.17, NeuroBehavioral Systems), while EEG was recorded from 62 channels (Ag/AgCl scalp ring electrodes, Easycap ®) according to the 10-10 international system using a 64-channel amplifier (BrainAmp, Brain Products ®). Additionally, electrooculogram (EOG) was recorded from two channels placed at the left outer canthus and above the left eye to control for horizontal and vertical eye movements, respectively. The ground electrode was placed at Fpz. Depending on the Covid-19 restrictions, the data was referenced either to the tip of the nose or to both ear lobes. As all data was re-referenced to common average later in the preprocessing and transformed into source space, the difference in reference should not be noticeable. The signal was recorded at a sampling rate of 1000 Hz, and impedances were kept below 10 kΩ.

After the EEG recordings, participants filled out a survey of how they solved the tasks and how much time they needed to perform the mental rankings.

## Behavioral data analyses and statistics
This study is not pre-registered. The log files from Presentation ® were imported to Excel and descriptive data was extracted. Reaction times were calculated from the onset of the probe to the response (≤4 s) and included

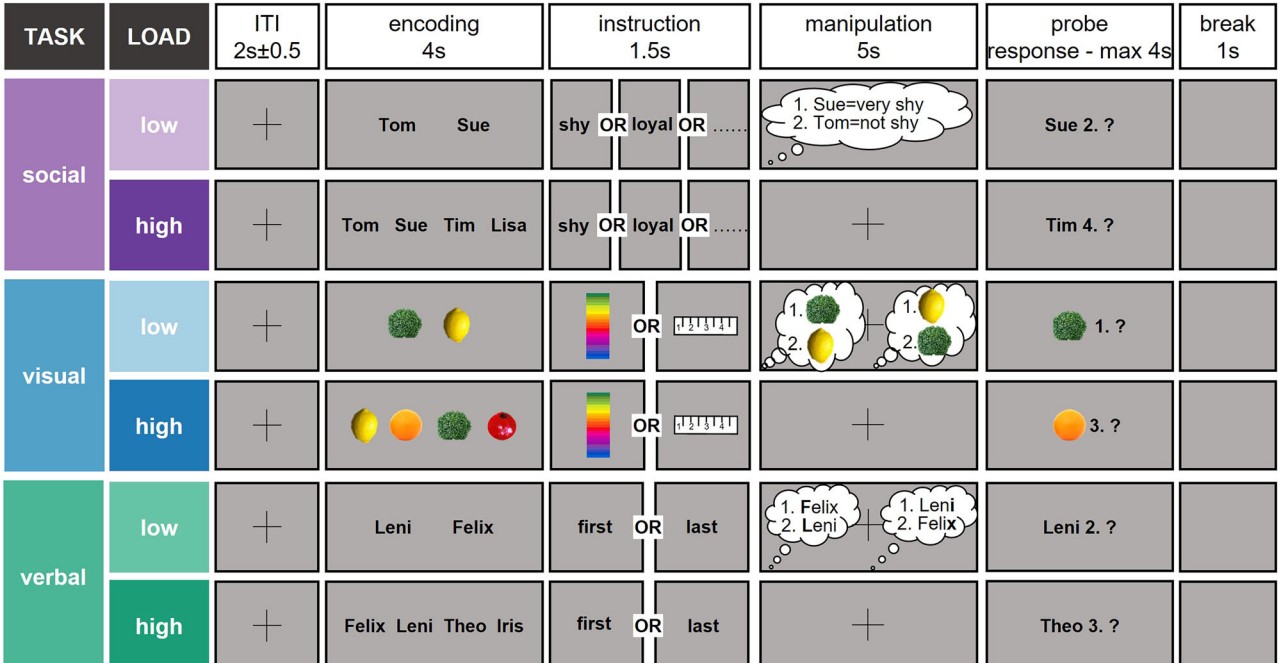

**Fig. 2 | Experimental design.** The experiment included three working memory tasks (social, visual and verbal), each consisting of a low and a high load condition. After an inter-trial interval (ITI) jittered around 2 s (± 0.5 s), either two (low load) or four (high load) stimuli were presented for 4 s for the participants to encode. In the social task[10,12], the displayed names were drawn from an individualized questionnaire of the screening (i.e., social questionnaire), in which participants rated their personal contacts according to traits. In the visual task, pictures of fruits and vegetables were displayed (pictures adapted from pexels.com and pixabay.com) and in the verbal task, German names were shown. Then the instruction was displayed for 1.5 s: In the social task, participants should mentally rank their personal contacts according to the displayed trait. In the visual task, the objects should be mentally sorted either according to color or size. In the verbal task[10], the names should be alphabetically sorted either according to their first or last letter. After the 5-s manipulation period, a probe asked whether a specific stimulus was at a specific position in the mental ranking. The participants pressed yes or no with their right hand as fast and accurately as possible on the left and right mouse key, respectively. Match and non-match probes were randomized and occurred equally often. After the response or maximally after 4 s, there was a 1-s break before the next trial started.

correct and incorrect responses. Accuracy in percentage was calculated by the number of correct responses divided by the total number of trials and multiplied with 100.

All 100 participants achieved a mean performance over all tasks significantly above chance level in the low ($M = 89\%$, SD = 5.58) and high ($M = 78\%$, SD = 6.63) load condition[83]. However, some of the participants indicated after the EEG session that they reversed the given order of manipulation in a block (e.g., sorted the objects from big to small instead of from small to big in the visual task). Indeed, six participants reversed the order of a task as they performed significantly below chance level in at least one block (<36% for the low and <13% for the high load condition). Thus, these six participants had to be excluded from behavioral data analyses. As the order of the ranking does not affect brain activity, we did not exclude these participants from the EEG analyses.

For statistical analyses, we calculated repeated-measures ANOVAs for accuracy (in %) and reaction time (in ms) with the within-subject factors task x load and the between-subject factor autistic-traits group ($N_{low\ AQ} = 48$, $N_{high\ AQ} = 46$) using IBM SPSS Statistics (version 29) and JASP 0.19.3. Possible violations of the normal distribution may bias ANOVA results—but there are indications that such violations may be acceptable[84,85]. Sphericity was confirmed for reaction times, accuracy had to be Greenhouse–Geisser corrected. Pairwise comparisons were performed based on estimated marginal means and Bonferroni-corrected. We calculated the planned pairwise comparisons one-tailed if we had a directed hypothesis and two-tailed if we assumed no differences between the autistic-traits groups: According to our hypothesis, we expected no differences between the autistic-traits groups in the low load condition (i.e., two-tailed testing), but a decrease in performance in the high load condition in the high autistic-traits group in comparison to the low autistic-traits group (one-tailed). We did not expect a difference between the autistic-traits groups in

the verbal task (two-tailed) but expected a decrease in performance in the visual and social tasks in the high autistic-traits group in comparison to the low autistic-traits group (one-tailed). The two-tailed 95%-confidence intervals (i.e., CI) of the effect size partial $\eta^2$ were approximated based on the $F$-value and degrees of freedom[86]. Behavioral data were visualized using MATLAB (R2016a).

**EEG data analyses and statistics**

**EEG preprocessing.** The continuous EEG data were preprocessed with BrainVision Analyzer 2.1 (Brain Products ®) to reduce and to control for artifacts as follows: In a visual inspection, large artifacts were removed from the data and if necessary, noisy channels were interpolated. Then the data were filtered between 0.1 and 100 Hz (48 dB/oct) and with a notch filter of 50 Hz. The EEG channels were re-referenced to a common average reference. Then an Independent Component Analysis (i.e., ICA) for ocular correction was applied in order to remove horizontal (saccades) and vertical (blinks) eye movements. Additionally, noise on certain channels could be removed using ICA. ICA components were selected based on visual inspection of scalp maps. On average, 11 components (SD = 4.5) were removed per participant. Channels that were still too noisy for analyses after ICA were interpolated (in total, one channel was interpolated for 3 participants and 3 channels were interpolated for one participant). Last, the data were again visually inspected, and all remaining artifacts were removed. Data of two individuals had to be excluded given that their data quality were insufficient (i.e., data without artifacts were ≤15 trials per condition[87]. The continuous artifact-free data were then segmented in the 5-s manipulation periods (+250 ms in the beginning and in the end of the segment to counteract filter artifacts). The remaining 98 participants (49 participants per autistic-traits group)

had on average between 29 and 31 manipulation period-trials (SD = 1.8–3.0) per task and load condition.

*A priori* source space transformation. These segments were then transformed into source space with the build-in module Low-Resolution Electromagnetic Tomography Analysis[88,89] in BrainVision Analyzer. Virtual electrodes were placed based on coordinates reported in neuroimaging studies for the social, verbal, and visual working memory task[10,12,90,91]: We defined the DMPFC as frontal region for FM-theta phase extraction. We defined 11 posterior regions of interest (ROIs) for gamma amplitude extraction: left and right temporal pole, left and right temporo-parietal junction, left and right inferior parietal lobe, left and right intraparietal sulcus, left, right and medial precuneus/posterior cingulate cortex (see Supplementary Table 1). For each ROI, we defined all the voxels, that were found in relevant neuroimaging studies for the respective regions[10,12,90,91] (see Supplementary Table 1). Then, the vectorial mean of the current density vectors across all the voxels included in one ROI was computed. X, y and z components of the mean current density vector were extracted from this value and averaged.

Power spectra. Only for the power spectra (but not for the phase-amplitude coupling analysis), the data were re-sampled to 512 Hz. The data were extracted from the first 2.5-s of the manipulation period. We only used the first 2.5 s of the 5-s manipulation period because participants indicated in the survey that they usually finished their mental ranking in the first half of the manipulation period in the low load condition. Power spectra derived using the Fast Fourier Transform ($\mu V^2$) were computed from 1 to 256 Hz with a 1 Hz resolution using the full spectrum and a symmetric hanning window in BrainVision Analyzer.

Fitting oscillations & one over *f*. In order to determine if the FM-theta peak is a real oscillation, we divided the power spectrum into an aperiodic component (i.e., 1/*f* like characteristics) and in periodic components (i.e., peaks rising above the aperiod components) using the FOOOF toolbox[92] with a Matlab wrapper (Matlab R2016a) and Phyton version 3.5. For the frequency range of 1–15 Hz, we allowed an infinite number of peaks and defined the peak width between 2 and 12 Hz, the minimal peak height with 0.1 $\mu V^2$ and peak threshold with 1.0 standard deviation of the aperiodic-removed power spectrum. We used a fixed aperiodic mode as our frequency range was rather small.

Complex Morlet wavelet transformation. 7-cycle complex Morlet wavelet transformation was applied. For the instantaneous gamma amplitude, data were filtered between 30–70 Hz in 10 Hz steps (30 Hz (26–34 Hz gauss), 40 Hz (34–46 Hz), 50 Hz (43–57 Hz), 60 Hz (51–69 Hz), 70 Hz (60–80 Hz). As control analysis, also theta amplitude was extracted for the 4–7 Hz frequency range. For FM-theta phase, 5.5 Hz center frequency was filtered with a Morlet parameter of 3.8, thus resulting in a 4–7 Hz frequency band. These complex wavelet coefficients were transformed to absolute phase values (-pi to +pi) with the *complex data measures solution* module. After gamma amplitude and FM-theta phase extraction, single 5-s manipulation period-trials were exported from BrainVision Analyzer.

Statistical analyses of amplitudes. As control analysis, we calculated repeated-measures ANOVAs per task and frequency band for the amplitudes of the extracted (complex morlet wavelet transformation) frequency bands of the first 2.5 s of the 5-s manipulation period using IBM SPSS Statistics (version 29). For FM-theta, we used the amplitude from the DMPFC as dependent variable and the within-subject factor load and between-subject factor autistic-traits group. For the gamma amplitudes, we calculated the ANOVAs for the within-subject factors load and posterior ROIs and between-subject factor autistic-traits group. Again, possible violations of the normal distribution may bias ANOVA

results—but there are indications that such violations may be acceptable[84,85]. If the sphericity was violated, we used the Greenhouse–Geisser correction.

Phase-amplitude coupling. The data were further analyzed in Matlab (R2016a) using in-lab scripts[23,93]. The posterior gamma amplitudes were z-transformed. The first 2.5-s of all manipulation period-trials were merged. As described above, we only used the first 2.5 s of the 5-s manipulation period because participants indicated in the survey that they usually finished their mental ranking in the first half of the manipulation period in the low load condition. Then, the z-transformed gamma amplitudes were sorted according to FM-theta phase angles (see Fig. 3) and averaged into 10 theta phase segments in order to reduce data complexity. As the values were very small, we multiplied the data by 10,000 for analyses, so information was not lost due to rounding decimal places and a second time by 10,000 for the figures for better illustration purposes. For the circular plots, the data were transformed to percentage values in order to have the same scale across tasks and conditions.

Hierarchical regression models. The z-transformed sorted gamma amplitudes were then further investigated by means of hierarchical regression analysis conducted in R (version 4.2.2, R Studio 2022.07.1)[94]. Hereby, possible violations of the normal distribution may bias the results, but there are indications that it should have little impact on the robustness of regression coefficient estimation[95,96].

We analyzed models for each task and gamma frequency band separately with posterior gamma amplitude as criterion variable and FM-theta phase segments, load, *a priori* defined posterior ROIs, and autistic-traits groups as predictor variables with random intercepts for each individual. We were predominantly interested in models comprising any interaction between FM-theta phase segments and load, given that we hypothesized that FM-theta phase modulated posterior gamma amplitude depending on load.

The significance of an interaction was determined by comparing the second-order Aikaike information criterion (AICc) value of an interaction model to the AICc value of a model comprising only respective main effect terms. The smaller the AICc value of a model, the more variance of its criterion can be explained by its predictor variables. Thus, a positive difference in AICc values between an interaction and a main effect model indicates that the interaction model explains the criterion variable better than the main effect model. We only analyzed interaction models further that achieved an AICc >10 indicating a large effect[97], and if the AICc value of the corresponding main effect models was 10 units smaller in comparison to a regression model without predictors and with random intercepts for each individual. We chose these conservative criteria in order to identify very strong and robust effects, i.e., to ensure that effects are true and not a product of testing multiple variables in complex interactions and separate regression models for tasks and frequency bands. It is important to note that AICc value comparisons reflect how well the data fit to a model and *not* whether some factor levels differ significantly from 0. Also, post-hoc analyses of any predictor level differences would not have been reasonable in this regard. Imagine data following a cosine curve: the largest difference would be obviously between baseline phase and peak values, but whether these differences were statistically different from null was less of interest for our investigation than whether the data followed a cosine pattern. Therefore, we applied a model fit comparison approach. Following this logic, we found that posterior gamma amplitude was best explained by a model with FM-theta phase segments interacting with (that means depending on) the load condition, the autistic-traits group (N = 49 participants per autistic-traits group), the specific gamma band and the task. The predictor posterior ROI did not significantly contribute to explaining variance in gamma amplitude. In other words, gamma amplitude and theta phase associations were statistically the same irrespective of the posterior ROI where gamma activity had been extracted. Thus, gamma amplitude was averaged over the posterior ROIs.

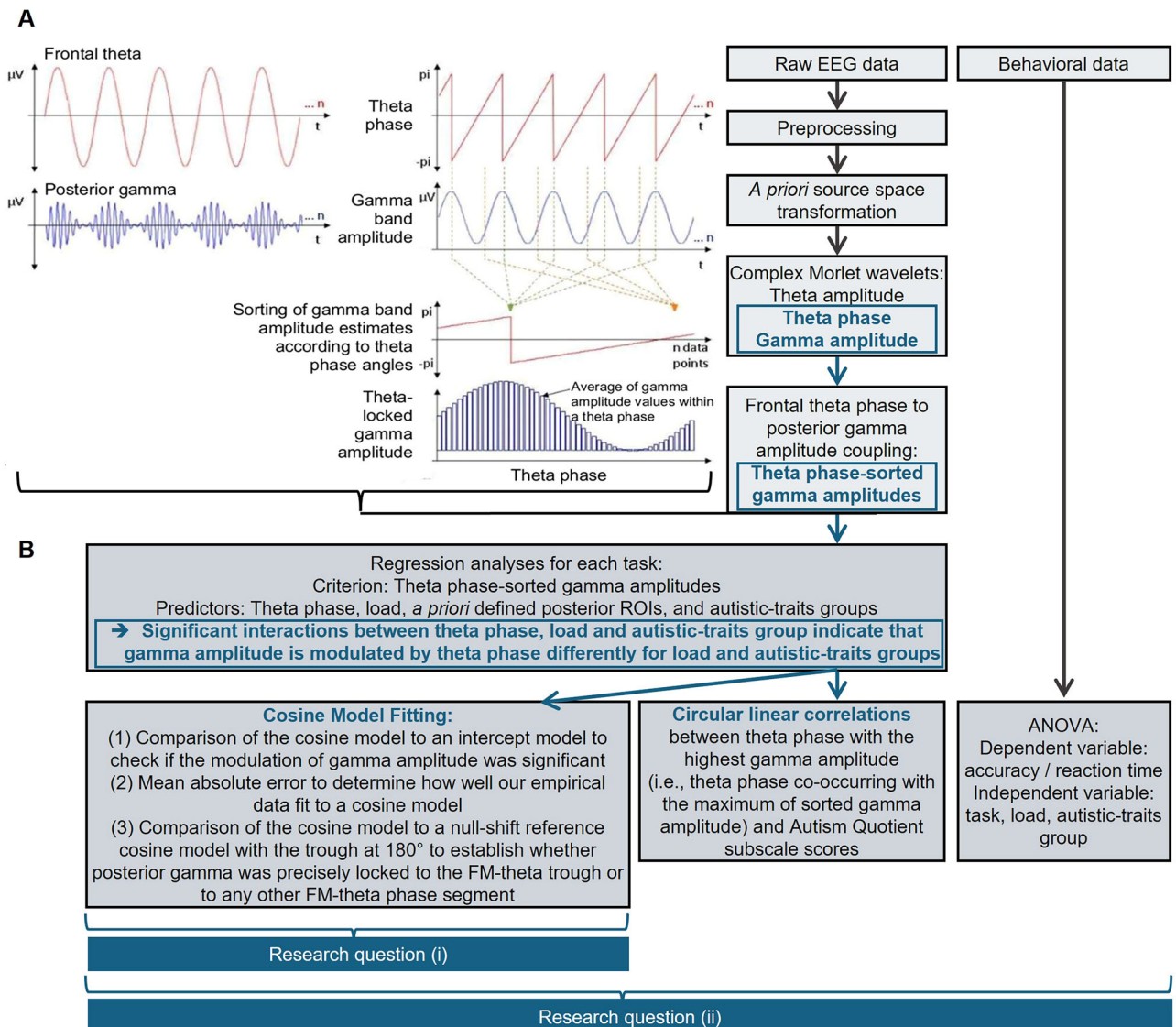

**Fig. 3 | Analysis pipeline. A** displays how the phase-amplitude coupling was computed: Using complex Morlet wavelet transformation, FM-theta absolute phase values (-pi to +pi) were extracted from the dorsomedial prefrontal cortex (DMPFC) and instantaneous gamma band amplitude values were extracted from 11 posterior ROIs (see Supplementary Table 1). Then the posterior gamma amplitudes were z-transformed and sorted according to the FM-theta phase angles. This resulted in averaged posterior gamma amplitude values related to different FM-theta phase angles. **B** illustrates the performed analyses with the theta phase-sorted gamma amplitudes and which of the two research questions they addressed. The combination of regression analyses and fitting steps (1) and (2) of the cosine model fitting can be interpreted as a measure of coupling magnitude: If there is no difference in coupling magnitude between conditions, the regression will not yield significant results. However, a significant regression does not necessarily imply a difference in phase-amplitude coupling magnitude, as the cosine model fitting steps (1) and (2) need to verify whether the coupling truly follows the theta phase. Step (3) of the cosine model fitting determines the phase shift of the coupling.

**Cosine-model fitting.** For the significant regression models (as determined based on AICc value differences), we investigated whether there was a cosine relationship between the average posterior gamma amplitude and FM-theta phase segments as described in Eq. 1. Hereby, $Y$ refers to individuals' average posterior gamma amplitude, $X$ to their FM-theta phase segment, $\alpha$, $\beta$, $\gamma$, and $\delta$ to free cosine model parameter values denoting the cosine amplitude, frequency shift, phase shift and off-set, respectively. We set $\beta$ to 1 to prevent an algorithmic solution neglecting our pre-defined FM-theta frequency. Then, $\alpha$, $\gamma$, and $\delta$ parameter values were estimated using a Gauss–Newton fitting algorithm.

$$Y = \alpha \times \cos\big(\beta \times (X + \gamma)\big) + \delta \qquad (1)$$

We analyzed our cosine model in three steps using R (version 4.2.2, R Studio 2022.07.1)[94]:

(1). We compared the AICc value of the cosine model fit to an intercept model (i.e., the average over all data points – which should be around zero). While positive values already indicated that the cosine model tends to fit better, AICc >2 indicates that the average posterior gamma amplitude estimates fit better to a cosine model and did not just randomly fluctuate around 0[97].

(2). The model fit to the cosine models was determined by the mean absolute error (MAE). It describes the mean absolute deviation between predicted average posterior gamma amplitude values based on cosine model parameter estimates $\hat{y}$ and individuals' empirical average posterior gamma amplitude values $y$ for each FM-theta phase segment. The smaller the error, the better the model fit. It would not be reasonable to provide the confidence interval as a marker of significance here as 0 could not be included in a very narrow confidence interval (indicating a good fit), while it might be likely included in a larger confidence interval (which would indicate a bad

model fit). Therefore, we will report the confidence interval range (i.e., MAEr): The smaller the MAEr, the better the model fit.

(3). We determined the most plausible cosine phase-shifts between regression models of differential load and autistic-traits groups. Therefore, we computed the range of most plausible γ estimates by means of two-tailed confidence intervals (i.e., CI) with a significance level of 5%. If 0° was not among the most plausible phase shift values, individuals' gamma amplitude was likely shifted, i.e., statistically different from a null-shift reference model. If 0° was included in the confidence interval, the highest gamma amplitude was locked in the trough of FM-theta (i.e., 180°), i.e., resembled the null-shift reference model.

The data were visualized using MATLAB (R2016a) and R ggplot2[98].

**Circular-linear correlations.** Circular-linear correlations between the FM-theta phase segment with the highest gamma amplitude (i.e., instantaneous FM-theta phase co-occurring with the maximum of sorted gamma amplitude) and AQ subscale scores over all participants as well as performance were computed[93,99]. These correlations were only calculated for the tasks and frequency bands showing a significant phase-amplitude coupling (a total of 42 correlations). The confidence intervals were approximated based on linear correlations[100,101]. The confidence level was not adjusted to account for multiple testing. Therefore, significant outcomes should be considered with caution.

## Results
### DMPFC theta phase to posterior gamma amplitude coupling
As described in the method section, we sorted posterior gamma amplitude as a function of FM-theta phase. The sorted posterior gamma amplitude values were used as dependent variable in regression models for the different working memory tasks with FM-theta phase segments, load, posterior ROIs and autistic-traits groups as predictor variables. If gamma amplitude is equally distributed across FM-theta phases, this would indicate no association between posterior gamma amplitude and FM-theta phase. If, however, sorted gamma amplitude systematically varies across different FM-theta phases, it will suggest interaction between FM-theta phase segments and posterior gamma (see ref. 23 for details and Fig. 3). Thus, only significant interactions involving the predictor FM-theta phase were indicative of posterior gamma amplitude being modulated by theta phase segments and thus were taken into account.

Regression models on FM-theta sorted gamma amplitude indicated significant interactions between FM-theta phase segments, load and autistic-traits group for the social task (at 70 Hz: second-order Aikaike information criterion value (AICc) = 34.51; at 60 Hz: AICc = 20.84; note, AICc values higher than 10 are considered as significant, see methods section for more information) and the visual task (70 Hz: AICc = 17.66). These effects generalized across posterior ROIs. There was no effect for FM-theta phase segments x posterior ROIs (AICc = −109.90 to 2.99, n.s.), FM-theta phase segments x posterior ROIs x load (AICc = −249.16 to −131.95, n.s.), FM-theta phase segments x posterior ROIs x autistic-traits group (AICc = −232.44 to −126.13, n.s.) or FM-theta phase segments x posterior ROIs x load x autistic-traits group (AICc = −500.64 to −318.89, n.s.) in any of the gamma frequency bands. This result indicates that gamma amplitude was not statistically significant different modulated by FM-theta phase at the different posterior ROIs (i.e., from the mentalizing network as well as those from the working memory system). Since the model indicated that the factor "posterior ROIs" did not contribute meaningful information, retaining this factor in the model was not justified. This is why, for any further analyses, sorted gamma amplitudes were averaged across all posterior ROIs.

Regression analysis for the verbal task did not yield any significant effects (AICc = −28.38 to 8.81, n.s.).

Thus, the results from the regression models indicate that a statistically significant interactions between autistic-traits groups and load conditions are predictive of gamma amplitude associated with certain theta phases. In order to further characterize the nature of the significant modulation of FM-theta phase-sorted gamma amplitude, we fitted our sorted gamma amplitude values to a theta cosine model: (1) This cosine model was compared to an intercept model to check if the modulation of gamma amplitude was significant, separately for respective load condition and autistic-traits group (AICc model comparison cosine vs. intercept). (2) We computed the MAE in order to determine how well our empirical data fit to a cosine model and report the confidence interval range (MAEr). A good model fit indicates that posterior gamma amplitudes are periodically modulated by FM-theta phase. (3) The cosine model was compared to a null-shift reference cosine model with the trough at 180° to establish whether posterior gamma was precisely locked to the FM-theta trough or to any other FM-theta phase segment.

**Low autistic-traits participants show expected phase-amplitude coupling (Research question 1).** Low autistic-traits individuals' posterior 70-Hz gamma amplitude was periodically modulated by FM-theta phase, both under low and high load conditions in the social and visual tasks (Fig. 4C, D top rows). In support of this, their data fit significantly better to a cosine model than an intercept model (social task: $AICc_{low\ load}$ = 6.11, $AICc_{high\ load}$ = 15.16, visual task: $AICc_{low\ load}$ = 10.68, $AICc_{high\ load}$ = 14.68; AICc >2 indicates that the data fit significantly better to a cosine than an intercept model; see methods). Moreover, their averaged posterior 70-Hz gamma amplitude values barely deviated from the predicted cosine model values as indicated by the MAEr (social task: $MAEr_{low\ load}$ = 1.67, $MAEr_{high\ load}$ = 2.40; visual task: $MAEr_{low\ load}$ = 2.17, $MAEr_{high\ load}$ = 2.39; MAEr indicates the confidence interval range, the smaller the MAEr, the better the model fit; see methods).

In the low load condition, strongest gamma activity was significantly shifted away from the trough by on average 53.57° (CI = [27.57°, 79.57°]) in the social and by on average 57.39° (CI = [37.54°, 77.20°]) in the visual task. In the high load condition, there was neither a significant phase shift in the social (CI = [−2.37°, 28.57°]) nor visual task (CI = [−23.18°, 8.58°]). This indicates that the maximal gamma amplitude was precisely locked to the FM-theta trough (180°) in the high load condition. In the 60-Hz frequency band in the social task, the phase-amplitude coupling was similar but slightly weaker than in the described 70-Hz frequency band (see Supplementary Fig. 1).

**High autistic-traits participants show a different brain communication pattern (Research question 2).** In the visual task (Fig. 4D bottom row), high autistic-traits individuals' posterior 70 Hz gamma amplitude was periodically modulated by FM-theta phase both in low (AICc = 11.28, MAEr = 1.61) and high load conditions (AICc = 17.25, MAEr = 4.12). However, there was no significant phase shift in the low load condition (CI = [−9.89°, 28.45°]). In the high load condition of the visual task, high autistic-trait individuals' posterior gamma was slightly shifted away from the FM-theta trough (minimal but significant phase shift of 23.30°, CI = [9.47°, 37.11°]).

In the social task (Fig. 4C bottom row), in the low load condition, the high autistic-traits individuals' data fit better to the intercept model (AICc = −4.88, MAEr = 5.17), suggesting no significant periodical modulation of 70-Hz posterior gamma amplitude by FM-theta at all. In the high load condition, posterior gamma amplitude was periodically modulated (AICc = 8.88, MAEr = 5.84). However, the maximum of sorted gamma amplitude was shifted in the opposite direction than that of low autistic-trait individuals (i.e., on average by −37.39°, CI = [−59.40°, −15.38°]). A shift from the FM-theta trough in the direction of 90° (in contrast to a shift in the direction of 270° as expected based on ref. 23), might point to suboptimal exchange of information in the fronto-temporoparietal network. Again, phase-amplitude coupling was similar in the 60-Hz frequency band (see Supplementary Fig. 1).

**Circular-linear correlations between phase-amplitude coupling and personality traits (Research question 2).** We ran circular-linear correlations between individual instantaneous FM-theta phase and autistic-

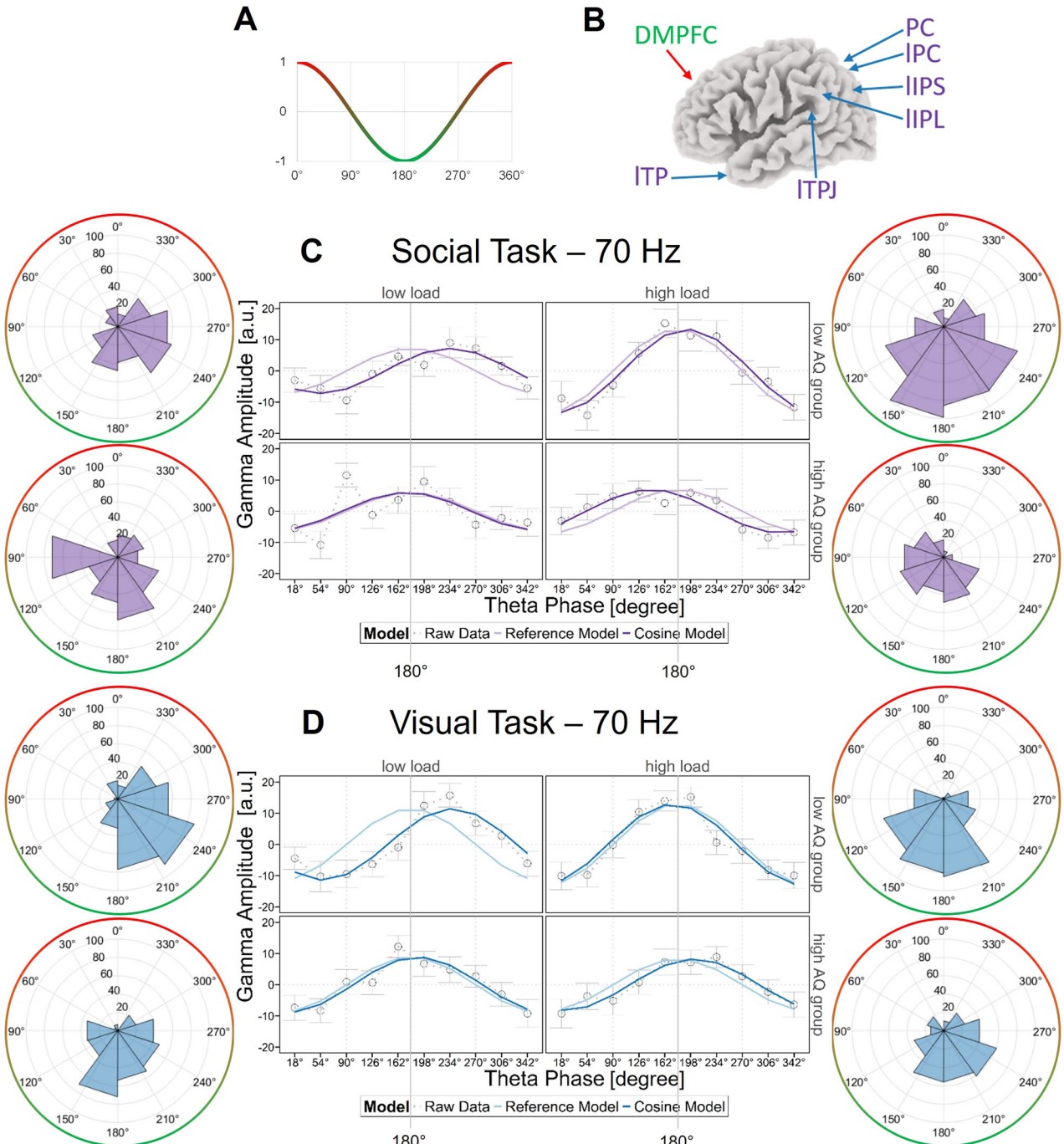

**Fig. 4 | Phase-amplitude coupling.** The phase-amplitude coupling for $N = 98$ participants is displayed. **A** The FM-theta trough is at 180° (green) and FM-theta peak is at 0°/360° (red) in our cosine model. **B** FM-theta phase was extracted from the dorsomedial prefrontal cortex (DMPFC; green/red). Posterior gamma amplitude (purple/blue) was extracted from 11 posterior regions of interest (ROIs): left/ right temporal pole (l/rTP), left/right temporo-parietal junction (l/rTPJ), left/right inferior parietal lobe (l/rIPL), left/right intraparietal sulcus (l/rIPS), left/right/medial precuneus/posterior cingulate cortex (l/rPC, PC). The arrows show the approximate left and medial ROIs, for all coordinates see Supplementary Table 1 (visualization of the brain derived from sLORETA[110]. **C, D** The z-transformed posterior instantaneous gamma amplitude was sorted according to instantaneous FM-theta phase and averaged over all 11 posterior ROIs. In the line charts, the grey dots indicate the mean of the empirical z-transformed and sorted 70-Hz gamma amplitudes, the grey whiskers indicate error bars (i.e., mean +/- standard error (SE)). The light purple/ blue lines indicate our null-shift reference cosine model (simulating that strongest gamma amplitudes were locked in the trough of FM-theta phase) and the dark

purple/blue lines the cosine model fitted to our empirical data. In the circular plots, z-transformed sorted gamma amplitudes (blue/purple) are displayed as percentage of the signal, FM-theta peak (0°) is indicated in red and FM-theta trough (180°) in green. In the **C** social (purple) and **D** visual (blue) tasks in the low autistic-traits (AQ) group (top rows), the highest gamma amplitudes were locked in the FM-theta trough (180°) in the high load (right) and shifted in the direction of 270° in the low load condition (left), demonstrating efficient and load dynamic coordination of fronto-temporoparietal networks. In the **C** social task in the high autistic-traits group (bottom row), there was no significant phase-amplitude coupling in the low load and a phase shift in the direction of 90° in the high load condition. This indicates inefficient coordination of working memory and mentalizing networks by FM-theta phase in high autistic-traits individuals. In the **D** visual task in the high autistic-traits group (bottom row), the highest gamma amplitudes were locked in the FM-theta trough (180°) already in the low load and slightly shifted in the high load condition, either pointing towards rigidity in working memory control or increased cognitive effort already in the low load condition in high autistic-traits individuals.

trait measures (AQ subscales) across participants. Individual, instantaneous FM-theta phase co-occurring with the maximum of sorted 60-Hz frequency band amplitude in the social task was correlated with the AQ subscales imagination in the low load condition ($r = 0.30$, $p = 0.014$, CI = [0.12, 0.48]) and with social skills in the high load condition ($r = 0.26$, $p = 0.036$, CI = [0.07, 0.45]). Thus, there is an indication that autistic personality traits may be related to dysregulated phase-amplitude coupling. However, these results should be considered with caution as the correlations were not corrected for multiple testing.

**Control analyses.** Theoretically, phase is independent of an EEG signal's amplitude. However, it is important to demonstrate that a signal used for phase-based coupling, such as FM-theta in our case, is truly oscillatory and not just an aperiodic signal. Thus, it was established whether there was oscillatory, periodic FM-theta present in the recorded EEG. We examined the power spectra for the time period for which phase-amplitude coupling had been analyzed previously. Within the DMPFC, a clear theta peak was visible for the low as well as for the high autistic-traits group (see Supplementary Fig. 2B). For both groups, one peak in the theta range with a power of 0.2 μV² above the aperiodic component (i.e., $1/f$ like characteristics) was extracted (FOOOF toolbox[92], model fit $r^2 > 0.9$ (error = 0.2)), confirming a periodic theta oscillation in the DMPFC[102]. The power spectra for posterior regions did not reveal any theta peak, showing that the theta oscillation was spatially specific to the DMPFC (see Supplementary Fig. 2C).

Second, we tested whether differences in amplitude could account for the obtained differences in phase-amplitude coupling between the groups. Extracting the amplitude at the DMPFC for the standard theta frequency band (4–7 Hz; based on ref. 23), we could furthermore show that FM-theta amplitudes were statistically not different between the autistic-traits groups (see Supplementary Table 2A). As expected, FM-theta amplitude was higher for the high load than low load condition[20,23]. It is also important to note that the reported phase-amplitude coupling effects were not driven by any general gamma amplitude differences between autistic-traits groups or load condition (see Supplementary Table 2B). We also did not obtain any significant interactions involving posterior ROIs with load or autistic-traits group for the 60- and 70-Hz gamma band in the social and visual tasks (see Supplementary Table 2B). Moreover, gamma amplitudes had been z-transformed for the phase-amplitude coupling, thus eliminating any possible absolute amplitude differences.

Third, we performed the cosine model fitting for the DMPFC theta phase to posterior gamma amplitude coupling separately for the posterior regions associated with the working memory system and with those associated with the mentalizing network (Supplementary Fig. 3). Both subfigures of the social and visual tasks in the Supplementary Fig. 3 show the phase–amplitude coupling pattern described in the main analysis (see Fig. 4). This is in line with the results of our regression models that gamma amplitude was not statistically significant different modulated by FM-theta phase at the different posterior ROIs.

Fourth, we tested whether evoked responses—although unlikely due to our rather lengthy analysis time segment of 2.5 s - could have had any impact on phase-amplitude coupling results[103]. Therefore, we shifted theta phase values and gamma amplitude by one trial, so that theta phase from trial 1 was now coupled with gamma amplitude from the last trial, and theta phase from trial 2 was coupled with gamma amplitude from the first trial. This way only stimulus evoked effects remained in the data without any induced effects surviving the re-alignment. We then calculated the same phase-amplitude coupling values, regression models, and cosine fitting models. The results indicated that - unlike in the original analyses - no significant interaction in the regression models between FM-theta phase segments, load and autistic-traits groups in the social task in the 60-Hz (AICc = −15.15, n.s.) and 70-Hz gamma amplitude (AICc = −33.84, n.s.) were found. In the visual task, a significant interaction was obtained in the 70-Hz gamma amplitude (AICc = 22.92). However, in contrast to the results from the original analysis, gamma amplitude was not significantly

periodically modulated by FM-theta or showed the reversed coupling pattern than in the main analyzes (see Supplementary Fig. 4). These results indicate that the findings from the main analyses cannot be explained by mere stimulus evoked effects.

### Left DLPFC theta phase to posterior gamma amplitude coupling
As a range of studies suggest that verbal working memory might rely more strongly on the left dorsolateral prefrontal cortex (lDLPFC)[10,32–35] than on the DMPFC, we analyzed the data with exactly the same procedure as reported above but used the left DLPFC instead of the DMPFC for theta phase extraction (see Supplementary Table 3). Regression models with theta phase segments from the left DLPFC, load and autistic-traits group as interaction terms suggest significant effects for 50-Hz posterior gamma amplitude in the verbal (AICc = 40.40) and visual (AICc = 28.07) tasks but not in the social task (AICc = −14.79 to −1.75, n.s.). However, the anticipated phase-amplitude coupling (as obtained with DMPFC theta activity for individuals with low autistic traits), could not be systematically found for the left DLPFC (see Supplementary Figs. 5, 6).

### Behavioral results (Research question 2)
**Load effects.** Task accuracy was higher (Fig. 5A; $F(1, 92) = 379.19$, $p < 0.001$, partial $\eta^2 = 0.81$, two-tailed CI = [0.73, 0.85]) and response times were shorter (Fig. 5C; $F(1, 92) = 802.17$, $p < 0.001$, partial $\eta^2 = 0.90$, two-tailed CI = [0.86, 0.92]) in the low than high memory load conditions across tasks and autistic-traits groups as expected (descriptive statistic in Supplementary Table 4). Importantly, this significant memory load modulation in performance was present in every task and comparable between tasks (Fig. 5).

**Task effects.** Accuracy was worse in the social compared to the non-social tasks (Fig. 5A, $F(1.67, 153.86) = 90.19$, $p < 0.001$, partial $\eta^2 = 0.50$, two-tailed CI = [0.38, 0.58]; as in ref. 10) but reaction times were faster in the social task (Fig. 5C; $F(2, 184) = 11.17$, $p < 0.001$, partial $\eta^2 = 0.11$, two-tailed CI = [0.03, 0.19]; as in ref. 90; descriptive statistic in Supplementary Table 4). This speed-accuracy trade off was reported by Meyer and colleagues[10,90] before. The difference in accuracy could be explained by differences in the reliability of the assessment criteria: The criteria for accuracy in the social task (based on an individualized questionnaire of the screening, i.e., the social questionnaire) were not as reliable but more subjective as in the visual (size and color of vegetables and fruits) and verbal (alphabet) tasks. The re-test reliability of the social questionnaire averaged over trait-items was $r = 0.74$ (CI = [0.67, 0.80]; see Supplementary Table 5). Importantly, the re-test reliability was comparable between autistic-traits groups (Supplementary Table 5) and thus cannot account for differences between tasks and between autistic-traits groups in performance reported below.

**Autistic-traits group effects.** We found two significant interactions involving the autistic-traits groups (descriptive statistic in Supplementary Table 4):

First, there was a significant interaction between the autistic-traits groups and the load conditions in reaction time (Fig. 5D; $F(1, 92) = 4.56$, $p = 0.035$, partial $\eta^2 = 0.05$, two-tailed CI = [0.0, 0.15]): There was no statistically significant difference in the reaction times between the autistic-traits groups in the low load condition ($M_{diff} = -10.95$, SE = 41.57, CI for $M_{diff} = [-93.52, 71.62]$, $t(92) = -0.26$, $p = 0.793$, two-tailed, Cohen's $d = -0.04$, two-tailed CI for $d = [-0.44, 0.36]$). There was a marginally significant statistic effect for shorter reaction times in individuals with low compared to high autistic traits in the high load condition ($M_{diff} = -91.31$, SE = 60.27, CI for $M_{diff} = [-211.00, 28.38]$, $t(92) = -1.51$, $p = 0.067$, one-tailed based on hypothesis: low < high autistic-traits group, Cohen's $d = -0.33$, two-tailed CI for $d = [-0.91, 0.26]$).

Second, for accuracy, there was a significant interaction between the autistic-traits groups and the tasks (Fig. 5B; $F(1.67, 153.86) = 3.46$, $p = 0.042$,

**Fig. 5 | Behavioral results.** The data distribution, interquartile range (dark vertical line), median (white dot) and mean (colored line) for $N = 96$ participants are displayed. Discontinuity markers were added as the $y$-axes do not originate at 0. **A** The low load (light colors) condition showed significantly higher accuracy than the high load (dark colors) condition. The social task (purple) showed significantly lower accuracy than the visual (blue) and verbal (turquoise) tasks. The significant main effects are indicated by the brackets. **B** There was a significant interaction between tasks and autistic-traits groups in accuracy. **C** The participants showed significantly shorter reaction times in the low load condition than in the high load condition. They showed significantly shorter reaction times in the social task than in the visual and verbal tasks. The significant main effects are indicated by the brackets. **D** There was a significant interaction between load conditions and the autistic-traits groups in reaction time.

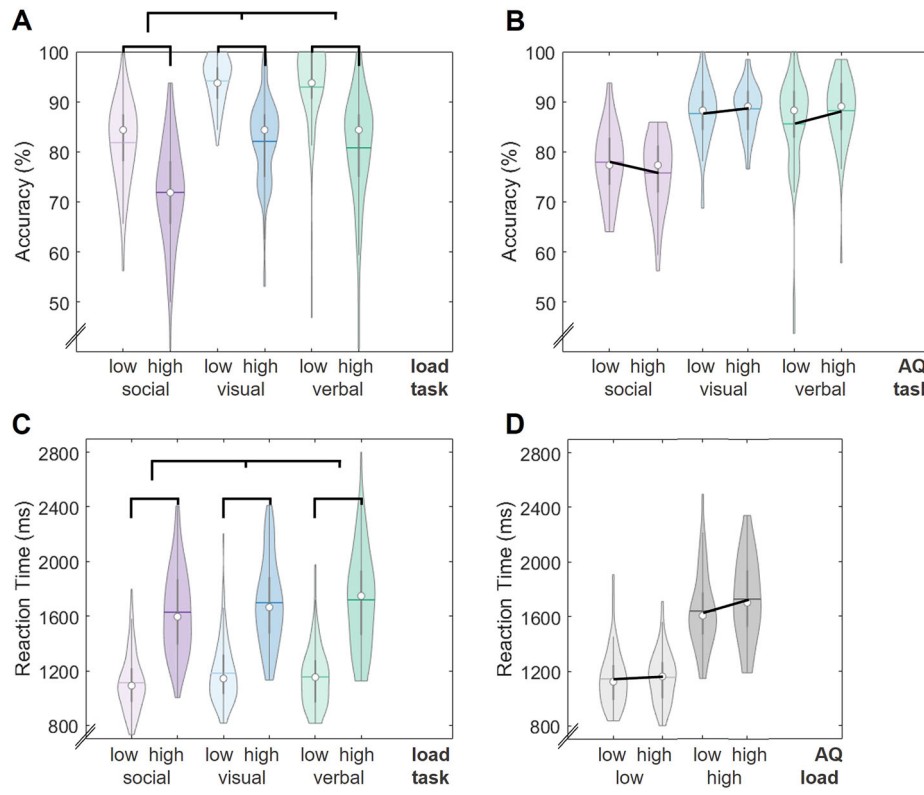

partial $\eta^2 = 0.04$, two-tailed CI = [0.0, 0.10]): Individuals with low - compared to high - autistic traits showed a marginally significant statistic effect to perform better in the social task ($M_{diff} = 2.15$, SE = 1.47, CI for $M_{diff} = [-0.77, 5.07]$, $t(92) = 1.46$, $p = 0.074$, one-tailed based on hypothesis: low > high autistic-traits group, Cohen's $d = 0.24$, two-tailed CI for $d = [-0.26, 0.75]$). There was no statistically significant difference in accuracy between the autistic-traits groups in the visual task ($M_{diff} = -0.92$, SE = 1.14, CI for $M_{diff} = [-3.20, 1.35]$, $t(92) = -0.81$, $p = 0.789$, one-tailed based on hypothesis: low > high autistic-traits group, Cohen's $d = -0.11$, two-tailed CI for $d = [-0.50, 0.29]$) and in the verbal task ($M_{diff} = -2.63$, SE = 1.88, CI for $M_{diff} = [-6.36, 1.09]$, $t(92) = -1.40$, $p = 0.164$, two-tailed, Cohen's $d = -0.30$, two-tailed CI for $d = [-0.95, 0.35]$).

## Discussion

Our results showed (i) how distant brain areas communicate with each other and how the prefrontal cortex distributes cognitive resources efficiently via the same phase-amplitude coupling mechanism in the social and visual working memory tasks. Moreover, (ii) the described phase-amplitude coupling mechanism could explain potential difficulties in coordinating cognitive processes and social functioning in individuals with high autistic personality traits.

### Low autistic-traits participants show unified visual working memory and mentalizing networks

Posterior working memory and mentalizing regions were co-activated and temporally coordinated by the identical prefrontal oscillatory mechanism during social as well as visual working memory tasks in individuals with low autistic personality traits: In the low autistic-traits group, posterior high frequency bursts were locked close to the trough of the FM-theta cycle in the high load and shifted towards the rising phase of the FM-theta cycle in the low load condition in both the social *and* the visual tasks.

This is conceptually reproducing a finding previously reported in young, non-clincial participants during a visuo-spatial working memory task[23]: Berger and colleagues found that in a demanding visual working memory task, posterior gamma was nested in the FM-theta trough, whereas

it was shifted towards the rising phase (towards the more inhibitory peak) in the easy visual working memory task.

Thus, posterior gamma amplitude is modulated by FM-theta phase so that optimal time windows for fronto-temporoparietal neural communications arise. Specifically, more effortful processing leads to more optimized modulation of posterior gamma activity by FM-theta phase. In easy, effortless processes to which fewer cognitive resources need to be allocated and, thus, where fronto-temporoparietal communication is not necessary or would even be a waste of neural resources, posterior gamma amplitude is aligned with the more inhibitory FM-theta phase, preventing unnecessary fronto-temporoparietal communication.

The current results show that FM-theta phase plays a vital role in coordinating distributed cortical networks during visual working memory processes and social cognition. The factor posterior ROI was not statistically significantly involved in any interactions in the regression analysis. Additionally, an exploratory supplemental analysis with the working memory system and the mentalizing network separated (Supplementary Fig. 3) showed that both subfigures (i.e., the one for the working memory system regions and the one for the mentalizing system regions) display the phase-amplitude coupling pattern described in the main analysis in low autistic-traits individuals (see Fig. 4). Thus, FM-theta phase does not only seem to control the working memory system but also the mentalizing network. As expected, the DMPFC constitutes a core region linking both systems[5,10,23,36,37,90,104], leading to our proposition of shared neural bases between the visual and social tasks.

In contrast, we could not find the above-described phase-amplitude coupling mechanism in the verbal task, neither for DMPFC, nor for left DLPFC theta phase. One explanation could be that even if the phase-amplitude mechanism extended to the left DLPFC, the applied methods might not have picked up the relevant signal for identifying such: Due to the anatomical structure of the cortex, more complex cortical folding in the lateral prefrontal regions might lead to dipole cancellation and increased variability across individuals. This could lead to problems in consistently detecting phase-amplitude coupling with surface EEG in the DLPFC. A study by Canolty and colleagues[21] supports this idea. Canolty et al. used

electrocorticographic (ECoG) recordings from the DLPFC while participants performed a range of cognitive tasks. ECoG is directly recorded from the cortical surface, which leads to a better signal-to-noise ratio and less dipole cancellation. And indeed, the authors report consistent phase-amplitude coupling between DLPFC-theta and gamma activity. Another important finding that Canolty and colleagues[21] reported is that similar cognitive tasks lead to similar phase-amplitude coupling patterns. In respect to the present study, this suggests that from a neural perspective the social cognition task and the visual working memory task were processed highly similarly. In contrast, the verbal working memory task might have been solved differently by the brain following Canolty and colleagues's rationale[21].

### Differences in autistic-traits groups could be explained by phase-amplitude coupling

As discussed above, only in the high load condition, low autistic-trait individuals tuned their fronto-posterior networks towards maximally efficient communication by precisely nesting posterior gamma into the trough of FM-theta (see ref. 23 for a discussion). The low load condition demanded less deployment of cognitive resources and less cognitive control, which meant that low autistic-trait individuals did not need to precisely nest posterior gamma into the FM-theta trough but could allow for a slight theta phase shift towards the rising phase. This dynamic regulation of cognitive control differed in high autistic-trait individuals: Already in the low load condition of the visual task, high autistic-trait participants displayed ultimate precision of FM-theta phase to gamma coupling; meaning that they might not be able to further increase cognitive control when the task becomes more difficult (i.e., in the high load condition).

This pattern might either reflect a certain rigidity of cognitive control functions, i.e., difficulties in load modulation[59,61,105], resulting in high autistic-traits individuals being stuck in the effortful mode and being unable to efficiently and dynamically couple and decouple the fronto-temporoparietal networks dependent on cognitive load (Fig. 4). Alternatively, this finding could indicate the expenditure of more cognitive effort for achieving performance comparable to the low autistic-traits group[47,56–58]. An observation rather speaking for the latter interpretation is that in the high load condition of the visual task, high autistic-trait individuals' posterior gamma was slightly shifted away from the FM-theta trough (Fig. 4). This might indicate that their optimal level of cognitive demand is closer to the low load condition, and that there is no optimal compensation mechanism for the high load condition[45,60,62].

This supports the idea of an inverted U shape response to working memory load[106]. These assumptions are also compatible with our obtained behavioral data, showing a significant interaction between load conditions and autistic-traits groups. In the pairwise comparisons with post-hoc correction, there was no statistically significant difference between reaction times in the low load and a marginally significant statistic effect for longer reaction times in the high load condition for high autistic-traits individuals in comparison to low autistic-traits participants (Fig. 5D).

In the social task, individuals with high autistic traits failed to show efficient coordination of fronto-temporoparietal networks by FM-theta phase to gamma coupling (Fig. 4). These findings of aberrant phase-amplitude coupling are in line with studies indicating that autistic individuals display lower activity in the prefrontal cortex and dysfunctional long-range connectivity, particularly fronto-posterior disconnectivity at lower frequencies[52–54]. Our described pattern of reduced FM-theta phase to gamma amplitude coupling in the social task is also compatible with our obtained behavioral data of a significant interaction between tasks and autistic-traits groups (see Fig. 5B).

Taken together, we found that individuals with high autistic personality traits have difficulties in dynamically adjusting fronto-temporoparietal communication. High autistic-traits individuals being stuck in the high effort mode in the visual task could potentially explain difficulties in working memory tasks reported especially at the high-demand level[45,60–62]. A failure of efficient neural coupling in the social task, in contrast, could explain major deficits in social abilities observed in individuals on the spectrum[41–43]. Thus,

it is not a binary mechanism (coupling = no difficulties vs. no coupling = difficulties) but depending on how this coupling and decoupling is dynamically used according to cognitive demands. The indication of a correlation between phase-amplitude coupling and autistic personality traits points in the direction of behavioral relevance of FM-theta phase to gamma amplitude coupling[55,107]. However, further research is required to determine the reliability of such effects.

### Limitations

Our study was conducted using EEG. Although EEG has a very good temporal resolution, its spatial resolution is rather poor, and the signals are only recorded at the surface of the head. We used 3D source reconstruction to estimate the signals at specific ROIs; however, this transformation is dependent on the implemented models. Intracranial recordings could better capture deep or spatially overlapping sources and magnetoencephalography could better detect signals originating from the sulci. Accordingly, such alternative recording methods might have been able to fully explain why we could find the phase-amplitude mechanism for DMPFC theta phase but not for left DLPFC theta phase for the verbal task.

Despite this limitation, we were able to conceptually reproduce - in a hypothesis-driven study with *a prior* defined ROIs and frequency bands - previous phase-amplitude coupling findings in the visual domain[23]. Our control analyses for FM-theta and posterior gamma amplitude confirm that differences between the autistic-traits groups really do rely on the phase-amplitude coupling mechanism and cannot be explained by evoked activity and mere differences in amplitude which could lead to differences in signal-to-noise ratio and accuracy of phase estimates[102].

Another limitation of the present study is that the obtained circular-linear correlations were not powerful enough to withstand multiple-testing corrections, despite our sample size of 100 participants. Additionally, the effect sizes of the significant interaction effects in our repeated-measures ANOVAs of our behavioral data were small. Thus, the confidence interval rounded to the second decimal appears to contain 0. This could be explained by the following: (1) there is no main effect of the autistic-traits group, so the interaction is mostly driven by one main effect (either load or task); (2) the confidence intervals for the effect size were only approximated based on F-values and degrees of freedom, which makes it more replicable but less sensitive. Generally, we cannot expect large effect sizes as the reported findings are based on the differences between two subclinical groups of participants who were carefully matched in sociodemographics, medical history and intelligence in order to mainly differ in the extent of autistic personality traits. This could be addressed by investigating our hypotheses in a sample covering the entire autism spectrum (including a larger portion of individuals with a clinical diagnosis). This would also allow us to investigate the effect of the AQ score as a metric predictor of inter-individual differences in behavior and theta-gamma phase-amplitude coupling.

Future research is necessary to pin down the causal relationship between the reported phase-amplitude coupling mechanism and deviations in social cognition and working memory processes. For instance, inhibiting the DMPFC with brain stimulation could reveal whether the phase-amplitude mechanism reported in this study is essential for dynamic allocation of cognitive resources. Moreover, the spatial specificity of the effects (i.e., whether they really can only be obtained for DMPFC) could potentially be addressed using a neurostimulation approach.

### Conclusions

This study showed that (i) effortful social cognition is coordinated by FM-theta phase to posterior gamma amplitude coupling, similar to visual working memory tasks. In both tasks, DMPFC theta phase coordinated neural communication with the working memory and mentalizing system. Moreover, this study included participants with a wide range of autistic personality traits - from no signs to a high degree of autistic personality traits - who were comparable in age, gender, education and intelligence. Strikingly, differences in the extent of autistic personality traits - while controlling for other confounding variables - were enough to manifest

themselves in different patterns of efficient coupling and decoupling of fronto-temporoparietal networks.

Thus, (ii) the here identified phase-amplitude coupling mechanism was sensitive enough to reveal differences between subclinical individuals depending on the extent of their autistic personality traits and suggested behavioral relevance for difficulties in effortful social cognition as well as high load conditions.

Moreover, this phase-amplitude coupling mechanism can be more or less fine-tuned, which can explain the different severity of deficits. Thus, the described phase-amplitude mechanism could inform clinical research and provide a unifying explanation for severe difficulties in social cognition and less severe difficulties in visual working memory in high-load conditions for individuals on the autism spectrum. It might not only be able to explain different characteristics within one mental health condition such as autism but also why these difficulties in cognition and social functioning often are shown in a variety of different mental health conditions. We might be able to generalize our findings to other psychiatric and neurological conditions - such as schizophrenia - to better understand problems in broad cognitive control and social cognition[108]. Recognizing this shared neural communication mechanism as reported in the current study could be a critical step toward understanding the underlying cause for individual differences in social cognition and working memory and informing cross-diagnostic applications for a range of mental health conditions.

## Data availability

The data supporting the findings of this study are made publicly available at the Open Science Framework (https://osf.io/pabfv/)[109].

## Code availability

The code used in this study are made publicly available at the Open Science Framework (https://osf.io/pabfv/)[109].

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

## Acknowledgements

We thank Doris Schmid for support with programming and piloting the experimental design and Daniela Gresch for support with EEG data preprocessing. We are very grateful for the many valuable discussions with Anna Lena Biel and Carola Romberg-Taylor in order to design the experimental tasks. We give our special thanks to Jörg von Mankowski for

providing tools to optimize our experimental design and advice during the complex analyses. We thank Eva Victoria Seegenschmiedt, Nele Habrecht and Ashley Yuan for assisting in designing the social questionnaire and the experimental tasks. This study was funded by a DFG grant to E.V.C.F. (FR 3961/1-1) and a SNSF Grant (10531F_220081) to P.S. The funders had no role in study design, data collection and analysis, decision to publish or preparation of the manuscript.

## Authors contributions

Conceptualization: E.V.C.F., P.S., Y.H.; Methodology: E.V.C.F., Y.H., P.S.; Software: E.V.C.F., Y.H., P.S.; Validation: E.V.C.F., Y.H., P.S.; Formal analysis: E.V.C.F., Y.H., P.S.; Investigation: E.V.C.F., Y.H., E.F.S., S.S.O., L.B.; Data curation and preprocessing: E.V.C.F., Y.H., L.B., E.F.S., S.S.O.; Resources: P.S., E.V.C.F.; Writing – original draft: E.V.C.F., Y.H., P.S.; Writing – review & editing: E.V.C.F., Y.H., P.S.; Visualization: E.V.C.F., Y.H., P.S.; Supervision: P.S., E.V.C.F.; Project administration: E.V.C.F.; Funding acquisition: E.V.C.F., P.S.

## Competing interests

The authors declare no competing interests.
