## [Transparent Peer Review file · Communications Psychology]

Theta-gamma phase amplitude coupling serves as a marker of social cognition and visual working memory deficits in individuals with elevated autistic traits

Corresponding Author: Professor Elisabeth Friedrich

Version 0:

Decision Letter:

Dear Professor Friedrich,

Thank you for your patience during the peer-review process. Your manuscript titled "Dysregulated neural communication as common signature for deficits in social cognition and working memory in high autistic trait individuals" has now been seen by 3 reviewers, whose comments are appended below. You will see that they find your work of some potential interest. However, they have raised quite substantial concerns that must be addressed. In light of these comments, we cannot accept the manuscript for publication, but would be interested in considering a revised version that fully addresses these serious concerns.

We hope you will find the Reviewers' comments useful as you decide how to proceed. Should additional work allow you to address these criticisms, we would be happy to look at a substantially revised manuscript. If you choose to take up this option, please highlight all changes in the manuscript text file, and provide a detailed point-by-point reply to the reviewers.

Editorially, we consider it crucial that the results are supported by rigorous phase-amplitude coupling analysis and statistical analyses in the revised manuscript. To this end, please address all methodological concerns raised by Reviewer #3, such as whether the stimulus-evoked response would potentially explain the phase-amplitude coupling results, and on the direct contrast comparing the phase-amplitude coupling results in the two groups with different autistic trait levels. Please also make sure that the motivation and the methodology of the study are clearly elaborated. As you address other presentational comments, please bear in mind that the paper needs to ultimately comply with Communications Psychology's reporting policies and formatting guidance, which you will find explained in the attached document and the formatting guide that is linked below.

Please ensure you follow our statistical guidelines when reporting statistics (<https://www.nature.com/commspsychol/submit/submission-guidelines#statistical-guidelines>). Please note in particular our requirements for the reporting and interpretation of null results. Non-significant findings derived from null-hypotheses significance tests should be reported in full, but may not be interpreted. Where you interpret null results, this interpretation must be based on Bayes Factors or equivalence tests.

I am attaching a checklist that details critical reporting requirements for the revised manuscript. Please attend to each item and ensure your manuscript is fully compliant. We are requesting that your manuscript aligns with these requirements as this facilitates the evaluation of your manuscript, reducing delays in re-review and potential future acceptance. If your revised manuscript is not aligned with these requests on major issues, such as those concerning statistics, it may be returned to you for further revisions without re-review. Additional information can be found in our style and formatting guide [Communciations Psychology formatting guide](https://www.nature.com/documents/commspsychol-style-formatting-guide-accept.pdf).

If the revision process takes significantly longer than five months, we will be happy to reconsider your paper at a later date,

provided it still presents a significant contribution to the literature at that stage.

Please use the following link to submit your

- revised manuscript,
- point-by-point response to the referees' comments,
- cover letter (as a separate document),
- the Editorial Policy Checklist (see below),
- the Reporting Summary (see below), and
- the completed Editorial Request Table (attached):

Link Redacted

Thank you for the opportunity to review your work.

Best regards,

Troy Lui

Troy Lui, PhD
Associate Editor
Communications Psychology

REVIEWER EXPERTISE:

Reviewer #1: cognition, EEG

Reviewer #2: autism, EEG

Reviewer #3: oscillatory analysis, EEG

REVIEWER REPORTS:

Reviewer #1 (Remarks to the Author):

In this research, the authors investigated the coordination of brain activity via slow oscillatory activity in the dorsomedial prefrontal cortex, as a shared mechanism involved in both working memory and mentalizing processes. Furthermore, the authors compare these processes across groups dichotomized based on low or high autistic traits. Overall, this is an interesting study with novel methodology. However, the manuscript could benefit from restructuring broadly to improve clarity, replicability, and impact. If permitted, I would recommend following the more traditional Introduction, Methods, Results, Discussion structure, as this manuscript is quite methodologically oriented, and it is currently challenging to interpret certain sections. Additional specific feedback is as follows:

Introduction:

- Generally, the Introduction would benefit from streamlining so that ideas flow from one to the next and provide a clearer rationale for the present study. Currently, while the authors summarize relevant literature and ideas, it is challenging to follow how these ideas are interconnected and set up the Methods used in the present study. The term phase-amplitude coupling is not mentioned at all in the Introduction.
- It might help to provide a more broad introduction to the EEG concepts – what they are and why they are relevant/important – in the Introduction to make the manuscript more accessible to a broader audience.
- The hypothetical in the opening is quite jarring. I would recommend a more straightforward explanation, but I understand that this may be a deliberate stylistic choice, so I leave it to the authors. However, I would suggest that if the authors choose to retain the hypothetical, they at least provide a more straightforward definition as well.
- Lines 90-94 (and throughout): It would be beneficial to expand on the studies cited (to summarize the methods, results) rather than provide sweeping statements. The language used should also be edited to ensure that it is appropriately academic and sensitive.
- Line 108 – The description here of Berger et al. (2019) is unclear.
- Line 126 – it would help the reader to provide the N for each group here, not just the total sample.

Results:

- Line 178 – The terminology used in the provided definition for ASD should be altered to meet existing academic standards. Citation(s) are all also missing.
- Currently, I find the rationale for dichotomizing the sample into high and low autistic-trait groups to be unclear. The authors

state that they decided not to separate participants into groups based on ASD diagnoses because of heterogeneity in ASD symptoms. However, they then dichotomous participants based on an Autism Quotient cut-off. Could the rationale for dichotomization be explained?

Discussion:

- Generally, the Discussion would benefit from editing throughout to ensure that language is clear and academic and that all concepts are properly explained and substantiated. For example:
 - o Line 447 – 451 – This should either be fleshed out more or deleted. Currently the claims are not clear and appear unsubstantiated.
 - o 495 – The findings of the cited study and how they are ‘exactly replicated’ should be elaborated on.
 - o Line 553 – rather than stating ‘one might speculate’, a proper discussion grounded in the cited literature would be appropriate.
- I am unsure why some statistics appear to be reported in the Discussion, rather than the Results.

Methods:

- Was any sociodemographic data beyond age/gender collected? This would be beneficial to report. Could it be clarified whether gender or sex was collected/reported?
- Were participants asked about/screened for color vision deficiency? It seems the color sorting task could be sensitive to such deficiencies. Furthermore, there is research suggesting that there is increased prevalence of color vision deficiencies in ASD.
- The phase-amplitude coupling section requires more detail. Currently the method used for calculating PAC is unclear to me and thus difficult to evaluate or replicate.
- Line 792 – what criteria were used to determine data quality was sufficient/insufficient?
- Line 625 – the relevance of autistic relatives or specialists in mathematics is unclear.
- Line 662 – it is unclear what is meant by one participant reporting mild depression in this context. How was this screened?
- Line 773 – It would help here to clarify which tests were hypothesis driven and tested with one-tailed tests. I think it would be more standard to use two-tailed tests in this context.
- Some other information in the Methods seems extraneous and could be streamlined to focus on the key scientific details.
 - o For example, what defines a comfortable chair? How was it determined that it was comfortable?

Other (minor):

- There are some general language errors throughout the manuscript that should be fixed before publication. Some of these, e.g., the use of ‘coherent’ instead of ‘consistent’ alter the meaning of sentences and thus could cause confusion.

Reviewer #2 (Remarks to the Author):

Dear Researchers

I deeply appreciate the opportunity to participate in the review of this research. I consider it to be an outstanding study that examines a broad sample of subjects using electroencephalography techniques. This approach is not only innovative but also explores traits related to the community, addressing potential conflicts linked to the field of mental health.

The work integrates behavioral aspects associated with cognitive functioning and intertwines them with actions and daily life behaviors, such as mentalization. Furthermore, it constitutes a compelling contribution to the field of neuroscience. However, certain aspects could be improved, which I have detailed in an attached file.

In the attached file, different questions have been answered and formulated to organize the information and establish a cohesive narrative that strengthens the presentation of this research. I hope these suggestions are taken into consideration with the aim of enhancing understanding among various readers and positioning this study as an innovative and relevant contribution to different communities.

Sincerely,

Reviewer #4 (Remarks to the Author):

This paper examines fronto-parietal phase-amplitude coupling (PAC) during three types of working memory tasks (visual, verbal, social) and how it depends on autistic traits. The authors focus on frontal theta phase and its modulation of parietal gamma power, a choice that is well motivated and described in detail. They report some differences in both the magnitude and phase of PAC between individuals divided into groups with low and high autistic scores, respectively.

The rationale behind the study is described clearly and corresponding hypotheses are tested with advanced statistical methods. I also appreciated the large sample size and thorough analysis of the data. However, I do have some concerns about the interpretation of the results and some methodological questions.

As stated in the title, the authors claim to show a “common signature for [...] social cognition and working memory”, and how it is changed in the presence of strong autistic traits. Apart from the title, this claim appears throughout the manuscript, e.g., “FM-theta phase does not only seem to control the fronto-parietal working memory system but also the mentalizing network”, “shared neural bases between the visual and social tasks”, etc. This claim puzzles me as the three tasks are all variants of a working memory task. How do the authors’ results demonstrate a functional role of fronto-parietal PAC that goes beyond

working memory? How can the “mentalizing network” be identified in the data? I think the manuscript would benefit from a more precise description of the cognitive functions the three tasks are designed to test, and what the obtained results mean in this respect.

It also remains unclear to me how the authors interpret the null effect in the verbal task. If fronto-parietal PAC underlies social and visual working memory, but not verbal working memory, what does that mean for its role in the brain? I appreciate the explanation that the authors propose in the Discussion, but it doesn't seem to be confirmed in the additional analysis they describe. I'm therefore missing a conclusion of whether PAC is a general mechanism (present for more than one task) or a specific one (absent in one task). Irrespective of this conclusion, I would find it helpful for the reader to see results for PAC in the verbal memory task as well.

The authors' analysis relies heavily on source-localised EEG data without the use of individual anatomical scans to improve the spatial accuracy (if I understand correctly). Is there any way the authors can estimate how well the DMPFC was localised? I wonder whether this could be addressed by showing that the observed theta-gamma PAC is specific for the ROIs selected and does not occur between other regions.

Results:

“regression models for the different working memory tasks with FM-theta phase segments [...] as predictor variables”

Could the authors please provide more details on the regression model used? It is unclear to me how this model includes theta phase. Is it used as a circular predictor in a linear regression model?

“[...] was similarly modulated by FM-theta phase”

I might have misunderstood something but I'm missing the effect of theta phase in the description of the statistical results (i.e. theta phase predicting gamma power).

Fig. 3: Would the authors please include a measure of between-subject variance?

Currently data from the two subject groups with high and low autistic traits seem to be analysed separately, with the exception of the circular-linear correlation. Given the authors' conclusion of differences in PAC between groups, a direct contrast between the two (predictive power of theta phase in the regression model, statistical contrast of phase etc) seems important to draw this conclusion.

[...] “these results should be considered with caution as the correlations were not corrected for multiple testing”. Could the authors please specify how many tests were made and/or include statistics for a version corrected for multiple comparisons?

PAC can also arise from stimulus-evoked responses and is influenced by their shape (Lozano-Soldevilla et al., 2016, *Frontiers*). Can the authors rule out such an effect in their data, e.g., spurious PAC produced by responses evoked by instruction offset that differ between conditions? Given that the interval analysed is 2.5 s long, this seems relatively unlikely, but it would be worth showing how these evoked responses look like for the different conditions.

The authors describe two types of changes in PAC: A phase shift and a change in magnitude. I think the manuscript would benefit from a clearer distinction of the two effects in the description of the results, and a speculation of how the two might differ in their functional significance.

Title: I suggest to avoid the term “communication” as it is not shown in the data.

“i.e. roughly 5 theta cycles / s x 2.5 s manipulation period x 30 trials = 375 data points per phase angle, resulting in over 13 000 data points per analyzed phase segment). This number of data points ensures a robust analysis even with a relatively low number of trials.”

I did not fully understand the purpose of this sentence. As the authors use 7-cycle wavelets, the 5 theta cycles do not seem treated independently from each other. It is also unclear what the 13000 data points mean, is it the number of EEG sample points analysed? I suggest to either provide more details or remove the two sentences altogether.

EDITORIAL POLICIES

We ask that you ensure your manuscript complies with our editorial policies and reporting requirements.

To that end, we require revised manuscripts to be accompanied by two completed items: a reporting summary that collects information on study design and procedure, and an editorial policy checklist that verifies compliance with all required editorial policies

- <https://www.nature.com/documents/nr-reporting-summary.zip>>Nature Research Reporting Summary
- <https://www.nature.com/documents/nr-editorial-policy-checklist.pdf>>Editorial Policy Checklist

All points on the policy checklist must be addressed. Your revised manuscript can only be sent back to the referees if these checklists are completed and uploaded with the revision.

Notes: If you have submitted a Stage 1 Registered Report, Review, Primer, Comment, or Perspective you do not need to submit these forms. If you have already submitted these forms, you may disregard this request.

Version 1:

Decision Letter:

Dear Professor Friedrich,

Thank you for your patience during the peer-review process. Your manuscript titled "Dysregulated Control of Neural Communication: A Common Marker of Social Cognition and Visual Working Memory Deficits in Individuals with Elevated Autistic Traits" has now been seen by 3 reviewers, and I include their comments at the end of this message. They find your work of interest but raised some important points. We are interested in the possibility of publishing your study in Communications Psychology, but would like to consider your responses to these concerns and assess a revised manuscript before we make a final decision on publication.

We therefore invite you to revise and resubmit your manuscript, along with a point-by-point response to the reviewers. Please highlight all changes in the manuscript text file.

Editorially, we consider it crucial that Reviewer #1's remaining concerns regarding the definition of autism and the analyses are thoroughly addressed in the revised manuscript.

I am attaching an Editorial Requests Table that details critical reporting requirements for the revised manuscript. Please attend to each item and ensure your manuscript is fully compliant. If your revised manuscript is not aligned with these requests on major issues, such as those concerning statistics, it may be returned to you for further revisions without re-review.

Please submit the following items:

- Revised manuscript

- Point-by-point response to the referees' comments
- Cover letter (as a separate document)
- [Nature Research Reporting Summary](https://www.nature.com/documents/nr-reporting-summary.pdf)
- Completed Editorial Request Table (attached).

via this link: Link Redacted .

Additional guidance is available in our style and formatting guide [Communications Psychology formatting guide](https://www.nature.com/documents/commpsychol-style-formatting-guide-accept.pdf).

Best regards,

Troy Lui

Troy Lui, PhD
Associate Editor
Communications Psychology

REVIEWER REPORTS:

Reviewer #1 (Remarks to the Author):

I appreciate the authors' responsiveness to reviewer feedback and revisions to the manuscript. The revised manuscript is substantively improved. However, I have several remaining concerns that should be addressed before the manuscript is suitable for publication.

- The hypothetical at the beginning is still confusing. I don't understand how it links into the key concepts in the paper. I'm particularly baffled by the leap from 'awkward work Christmas party' to psychiatric disorders. It's also not a culturally sensitive example.
- The revised definition of autism remains inaccurate and misleading. This needs to be fixed before publication. Furthermore, reading through the revised manuscript, I wonder if the authors are solely interested in autistic traits, and not ASD? Perhaps it would make sense to reframe the manuscript throughout to focus on autistic traits only.
- The authors stated that they 'adopted a combination of person-first and identity-first language in this paper to acknowledge and reflect the diverse preferences within the community.' However, after reading the paper, I only found one instance of identity-first language, with the paper otherwise written entirely with person-first language. Focusing on autistic traits would resolve this issue too.
- The rationale for dichotomizing participants provided by the authors in their response actually reads like more support for not dichotomizing participants. Based on this, I think it would make sense to use the AQ as a continuous measure, at least in supplementary analyses.
- The application of one-tailed post hoc tests in this context is not conventional. While further clarification has been provided about hypotheses that were directional, these contrasts weren't pre-registered and could inflate Type I error. Please at least report two-tailed results in the supplement.
- The authors report that all participants underwent a 'clinical interview,' but also state that the Beck Depression Inventory (BDI) was used to 'eliminate the possibility of depression' in one participant who 'reported sadness.' This is unclear for several reasons. First, sadness is a normal human emotion and does not indicate depression in the absence of other clinical symptoms. Second, the BDI is a self-report symptom inventory, not a diagnostic tool. If a diagnostic clinical interview was

conducted, it is unclear why the BDI would be needed afterward to confirm the absence of depression. The rationale for using the BDI in this context should be clarified.

- The revised text still does not explain why mathematics specialists are of particular relevance for comparison in this context or justify the AQ cut-offs. I suggest removing this entirely.
- The authors elaborated on Berger et al, which was helpful. However, they still using the term 'exactly replicating'. Is this an exact or conceptual replication?
- Was surrogate testing performed as part of the PAC analysis?
- The PAC calculation is non-conventional. Could a comparison with traditional methods be added to the supplement?

Reviewer #2 (Remarks to the Author):

I believe the changes made in response to the previous review have been beneficial, leading to improved analyses and a clearer overall structure of the manuscript. The article presents a novel approach that will certainly be of interest to the scientific community, particularly for its contribution to understanding the neurocognitive mechanisms associated with autistic traits. The conclusions appear original and well supported.

However, I would like to raise a point concerning the terminology used to refer to the studied population. In the current field of autism research, there is a growing tendency to use the term "autistic people" instead of "people with autism", in alignment with the preferences expressed by many individuals on the autism spectrum and their communities. While I understand that the study focuses on individuals with autistic traits rather than clinical diagnoses, I encourage the authors to explore this discussion further. If supported by recent literature, I suggest making a small but meaningful adjustment to the way the population is referred to in the manuscript, out of respect for the community being studied.

All other revisions have been adequately addressed. The methodology is now clearly structured, which enhances the reproducibility of the study. The systematization of procedures adds rigor and facilitates replication, which are essential qualities for research of this nature.

In my opinion, the manuscript is suitable for publication.

Reviewer #4 (Remarks to the Author):

Thank you for this thoroughly revised manuscript. My questions have been addressed.

If you experience problems in linking your ORCID, please contact the Platform Support Helpdesk.

Version 2:

Decision Letter:

Dear Professor Friedrich,

Your manuscript titled "Theta-gamma phase amplitude coupling serves as a marker of social cognition and visual working memory deficits in individuals with elevated autistic traits" has now been seen by our reviewers, whose comments appear below. In light of their advice I am delighted to say that we are happy, in principle, to publish a suitably revised version in Communications Psychology.

We therefore invite you to revise your paper one last time to address the remaining concerns of our reviewers and a list of editorial requests. At the same time we ask that you edit your manuscript to comply with our format requirements and to maximise the accessibility and therefore the impact of your work.

EDITORIAL REQUESTS:

Please note that we will not proceed with the publication of the article unless all remaining concerns are thoroughly addressed in the revised manuscript. Unless t-tests are planned comparisons, they must be two-tailed. In the present case, we note that the results from the one-tailed t-tests are non-significant, or marginally significant. By the journal's statistics guidelines, marginally significant tests may and should be reported, but cannot be interpreted. Therefore, regardless of whether these are planned comparisons that qualify for one-sided tests, or post-hoc analyses implemented to test the direction of interaction effects (which would need to be two-tailed), these findings may not be interpreted.

SUBMISSION INFORMATION:

OPEN ACCESS:

* DATA AVAILABILITY:

Link Redacted

Best regards,

Troy Lui

Troy Lui, PhD
Associate Editor
Communications Psychology

REVIEWERS' COMMENTS:

Reviewer #1 (Remarks to the Author):

The authors have again been responsive to feedback, and the revised manuscript is substantively improved. However, the authors & I remain at odds about the dichotomization of participant groups. The authors argue that dichotomization is inherent to the study design and that analyzing the AQ as a continuous variable "could be considered inconsistent with good scientific practice." Yet, throughout the manuscript, the authors frame their work within the conceptualization of autism as a spectrum and explicitly justify not using binary diagnostic categories on that basis. These positions seem difficult to reconcile. If the study's design truly precludes re-analysis using continuous AQ scores, the authors should at minimum acknowledge this design as a limitation.

Additionally, the authors' response regarding one- versus two-tailed tests does not adequately resolve this concern. As noted in my previous comments, the use of one-tailed post hoc tests in this context is unconventional and increases the risk of Type I error. As currently presented, the manuscript's reliance of selective one-tailed tests, as well as lack of correction for multiple comparisons, results in an anti-conservative analytic approach. Yet these findings are discussed mechanistically in the Discussion section. It cannot be assumed that the journal's broader readership will recognize this non-standard reporting, recalculate p values, and reinterpret the results accordingly. In my view, the prior suggestion to report results both one- and two-tailed was already a reasonable and transparent compromise.

Dear Reviewers and dear Editor,

Thank you for reviewing our manuscript. We revised it according to your valuable suggestions and believe that it has improved substantially.

Please find our changes in red in the manuscript and our answers to your suggestions below.

REVIEWER EXPERTISE:

Reviewer #1: cognition, EEG

Reviewer #2: autism, EEG

Reviewer #3: oscillatory analysis, EEG

REVIEWER REPORTS:

Reviewer #1 (Remarks to the Author):

In this research, the authors investigated the coordination of brain activity via slow oscillatory activity in the dorsomedial prefrontal cortex, as a shared mechanism involved in both working memory and mentalizing processes. Furthermore, the authors compare these processes across groups dichotomized based on low or high autistic traits. Overall, this is an interesting study with novel methodology. However, the manuscript could benefit from restructuring broadly to improve clarity, replicability, and impact. If permitted, I would recommend following the more traditional Introduction, Methods, Results, Discussion structure, as this manuscript is quite methodologically oriented, and it is currently challenging to interpret certain sections. Additional specific feedback is as follows:

Thank you for the valuable feedback, we re-structured our manuscript according to your suggestions and are confident that we improved clarity, replicability, and impact.

Introduction:

- Generally, the Introduction would benefit from streamlining so that ideas flow from one to the next and provide a clearer rationale for the present study. Currently, while the authors summarize relevant literature and ideas, it is challenging to follow how these ideas are interconnected and set up the Methods used in the present study. The term phase-amplitude coupling is not mentioned at all in the Introduction.

We re-arranged the introduction in a way that the flow of ideas is more understandable. Moreover, we added more information about our rationale. As we present the methods now before the results (according to your suggestion), we could also neglect information about the methods in the introduction and focus on the literature, research question and hypotheses instead. We also introduced the term “phase-amplitude coupling” in the introduction:

“Coordination of neural communication between distributed cortical regions during demanding visual working memory processes, was recently shown to be implemented via interregional phase-amplitude coupling^{23,24.}”

- It might help to provide a broader introduction to the EEG concepts – what they are and why they are relevant/important – in the Introduction to make the manuscript more accessible to a broader audience.

Thank you. We added some information about the role of FM-theta and coupling mechanisms. Due to length restrictions, we were unable to give a broader explanation of EEG rhythms in general but cited additional references who do:

“The dorsomedial prefrontal cortex (DMPFC) has been described as a hub region for social cognition¹⁰ as well as a key region for (non-social) working memory processing^{13–16}. This makes the DMPFC a prime candidate for linking the mentalizing network and the working memory system. And it is exactly the DMPFC which – under high cognitive demand – expresses slow (4 to 7 Hz), high amplitude rhythmical activity that can be measured using the human electroencephalogram (EEG). This so-called frontal-midline theta activity (FM-theta)^{13–16} has been associated with cognitively demanding working memory tasks and central executive functions^{17–20}. Moreover, FM-theta activity seems to play a key role in organizing and synchronizing brain activity across the cortex^{17,21} and, thus, in establishing effective neural communication²².“

- The hypothetical in the opening is quite jarring. I would recommend a more straightforward explanation, but I understand that this may be a deliberate stylistic choice, so I leave it to the authors. However, I would suggest that if the authors choose to retain the hypothetical, they at least provide a more straightforward definition as well.

We decided to leave the opening as it was really important to us and we believe it helps to understand the concepts better. However, we added a more straightforward definition to social cognition, mentalizing network and working memory:

“Difficulties with social cognition (i.e., cognitive processes needed to store social information and to apply it to oneself and others) can not only be challenging in daily life but they are also related to various psychiatric disorders¹.“

“Attributing mental traits to another person, such as what kind of presents someone might like or dislike, strongly relies on activity within the so-called mentalizing network in the brain^{2,3 4–6}. It includes a group of cortical regions strongly associated with social cognition in the dorsal parietal cortex, temporal cortex, cingulate cortex, and dorsal and medial prefrontal cortex^{2–6}. However, usually, high-level social cognitive functioning also requires flexible transient storage and utilization of large quantities of information – processing known as working memory⁷. Accordingly, in situations as described in the example above, one would also expect an involvement of the working memory system in addition to the mentalizing network. “

- Lines 90-94 (and throughout): It would be beneficial to expand on the studies cited (to summarize the methods, results) rather than provide sweeping statements. The language used should also be edited to ensure that it is appropriately academic and sensitive.

Thank you, we expanded the description of the cited studies and made sure to use academic and sensitive language as well as to acknowledge and reflect the diverse preferences within the community.

“Also coordinating mentalizing networks and working memory systems can be of particular importance in conditions with difficulty in social processing as well as transiently retaining and utilizing non-social information. This is the case in Autism Spectrum Disorder (ASD). For individuals with ASD ¹ mentalizing and reasoning about others can be particularly challenging ^{41–43}. In addition, difficulties in non-social working memory tasks ⁴⁴, especially in the visuo-spatial domain ^{45–51} can be observed in individuals with ASD. However, even in the sub-clinical, general population the extent of autistic traits predicts visual working memory performance ⁴⁶.

On the neural level, several studies found that individuals with ASD displayed lower activity in the prefrontal cortex and long-rang dysconnectivity, especially a frontal-posterior under-connectivity in lower frequencies ^{52–54}. Moreover, theta activity was found to be correlated with autistic symptomatology ⁵⁵: in neurotypical children theta activity was predictive for the amount of autistic behavior traits ⁵⁵.

While several studies indicate deviations in brain activity in individuals with ASD, there is a lack of evidence for behavioral effects in rather easy cognitive tasks, possibly pointing towards neural peculiarities in individuals with ASD partly merely reflecting efficient compensation mechanisms so that the same level of behavioral performance as in neurotypical individuals can be achieved ^{47,56,57}. However, in demanding cognitive tasks, this strategy seems to reach its limits and difficulties in adjusting to cognitively more effortful tasks become evident ^{45,58–61}.

¹ We have adopted a combination of person-first and identity-first language in this paper to acknowledge and reflect the diverse preferences within the community. (Buijsman et al., 2023; Taboas et al., 2023).”

• Line 108 – The description here of Berger et al. (2019) is unclear.

We rephrased the paragraph as follows to improve clarity:

“Coordination of neural communication between distributed cortical regions during demanding visual working memory processes, was recently shown to be implemented via interregional phase-amplitude coupling ^{23,24}: In this way, fast frequency posterior cortical activity (in the gamma frequency range - which can be considered a proxy for working memory activity ^{25–27}) is nested into specific phases of the FM-theta cycle ²³. Berger and colleagues showed that this phase-amplitude coupling mechanism was working memory load-dependent ²³. In a visual working memory task requiring a lot of cognitive control, posterior gamma activity was nested into FM-theta trough ²³– the excitatory phase ^{28–30}. In contrast, in an easy visual working memory task, posterior gamma was nested more towards the rather inhibitory peak phase. This way, depending on how effortful a working memory process might be and dependent on how much cognitive control is necessary, fronto-parietal cortical networks can either be coupled (gamma amplitude peak in theta trough) or actively dis-engaged (gamma amplitude peak shifted away from theta trough) to preserve cognitive resources. Applying this oscillatory coupling/decoupling mechanism, the DMPFC, therefore, can be considered perfectly adjusted for interfacing the mentalizing and the visual working memory networks ^{10,23}.”

- Line 126 – it would help the reader to provide the N for each group here, not just the total sample.

As we have re-structured the manuscript according to your comment so that the methods are before the results, we deleted this information in the introduction, as it is in the methods section and will be read before the results. We have also added Table 1 to illustrate the sample better.

Results:

- Line 178 – The terminology used in the provided definition for ASD should be altered to meet existing academic standards. Citation(s) are all also missing.

As we swapped methods and result section around, this chapter “Autistic Personality Traits” is now in the beginning of the Methods section:

We changed the sentence to the following and added a citation:

“Autistic Personality Traits

ASD is a spectrum disorder⁶⁵ with the extreme end of the spectrum considered clinically relevant . Nevertheless, the spectrum is present in the sub-clinical range as well. Since the tasks used in this study were cognitively highly demanding and required a certain level of social abilities, instead of comparing a clinical ASD group with controls, young adults who were comparable in age, gender, education, intellect and medical history but who differed in the extent of autistic personality traits (Table 1) were recruited.“

- Currently, I find the rationale for dichotomizing the sample into high and low autistic-trait groups to be unclear. The authors state that they decided not to separate participants into groups based on ASD diagnoses because of heterogeneity in ASD symptoms. However, they then dichotomous participants based on an Autism Quotient cut-off. Could the rationale for dichotomization be explained?

Thank you for this comment! Our rationale behind this was the following: Dividing participants into groups with and without a diagnosis would not allow us to match the groups as closely as we did because individuals with ASD could range in their symptoms from mild to severe. So, we might have a group showing great heterogeneity in intellectual abilities, verbal abilities, social contacts, medication intake and co-occurring psychiatric disorders, which would make it very hard to ascribe our results to autistic personality traits per se. And as we found literature that differences can already be found in subclinical individuals differing in the extent of autistic personality traits, we decided to recruit participants from the same student pool, match them closely in sociodemographics, medical history and intelligence and try to make the autistic personality trait the “only” difference between them.

We explain this rationale also in the chapter “Autistic Personality Traits” in the beginning of the methods section:

“Autistic Personality Traits

ASD is a spectrum disorder⁶⁵ with the extreme end of the spectrum considered clinically relevant . Nevertheless, the spectrum is present in the sub-clinical range as well. Since the tasks used in this study were cognitively highly demanding and required a certain level of

social abilities, instead of comparing a clinical ASD group with controls, young adults who were comparable in age, gender, education, intellect and medical history but who differed in the extent of autistic personality traits (Table 1) were recruited.“

Discussion:

- Generally, the Discussion would benefit from editing throughout to ensure that language is clear and academic and that all concepts are properly explained and substantiated. For example:

Thank you for this comment, we revised the discussion to improve its clarity. Moreover, we improved its structure to streamline our ideas.

- o Line 447 – 451 – This should either be fleshed out more or deleted. Currently the claims are not clear and appear unsubstantiated.

We deleted this paragraph as our manuscript pushes the length recommendation, and we agree that it is not essential for the paper.

- o 495 – The findings of the cited study and how they are ‘exactly replicated’ should be elaborated on.

We elaborated the study and added the following description in the discussion section:

“This is exactly replicating a finding previously reported in young, healthy participants during a visuo-spatial working memory task²³: Berger and colleagues found that in a demanding visual working memory task, posterior gamma was nested in the FM-theta trough, whereas it was shifted towards the rising phase (towards the more inhibitory peak) in the easy visual working memory task.

Thus, posterior gamma amplitude is modulated by FM-theta phase so that optimal time windows for fronto-parietal neural communications arise. Specifically, more effortful processing leads to more optimized modulation of posterior gamma activity by FM-theta phase. In easy, effortless processes to which fewer cognitive resources need to be allocated and, thus, where fronto-parietal communication is not necessary or would even be a waste of neural resources, posterior gamma amplitude is aligned with the more inhibitory FM-theta phase, preventing unnecessary fronto-parietal communication. “

- o Line 553 – rather than stating ‘one might speculate’, a proper discussion grounded in the cited literature would be appropriate.

We apologize for this phrase. This argumentation was routed in literature and we therefore changed it to:

“This supports the idea of an inverted U shape response to working memory load¹⁰². “

- I am unsure why some statistics appear to be reported in the Discussion, rather than the Results.

Thank you for this comment. We have moved the statistics from the discussion section to the result section in a new chapter: *“Control analyses “*.

Methods:

- Was any sociodemographic data beyond age/gender collected? This would be beneficial to report. Could it be clarified whether gender or sex was collected/reported?

Yes, we also recorded education and mother tongue as sociodemographic data. We revised the former Table 1 in order to make the recorded data clearer.

We recorded gender as indicated in Table 1 (gender: number of women/men) and indicated in the method section:

“Gender was determined based on information provided by the participants.”

- Were participants asked about/screened for color vision deficiency? It seems the color sorting task could be sensitive to such deficiencies. Furthermore, there is research suggesting that there is increased prevalence of color vision deficiencies in ASD.

Yes, we used the Ishihara color test to make sure all participants were able to see color. We made this information more prevalent in Table 1.

- The phase-amplitude coupling section requires more detail. Currently the method used for calculating PAC is unclear to me and thus difficult to evaluate or replicate.

We added a new figure in order to make our analyses more comprehensive. Please see Figure 2 for detailed explanation of our analyses pipeline.

- Line 792 – what criteria were used to determine data quality was sufficient/insufficient?

The EEG data preprocessing followed the described and standardized steps in order to make sure that the data does not contain artifacts. If after all these measures to get rid of artifacts the data still contained too many artifacts to be used and removing all the artifacts would have meant that not enough data is left (i.e., below 15 trials per condition), data quality was insufficient.

We added this in the manuscript:

“Data of two individuals had to be excluded given that their data quality was insufficient (i.e., data without artifacts was ≤ 15 trials per condition⁸⁵).

- Line 625 – the relevance of autistic relatives or specialists in mathematics is unclear.

Thank you for this comment. We cited studies investigating subclinical individuals with high autistic personality traits, so our AQ value can be put in the context of literature.

We deleted this statement from the “screening” section in the methods and elaborated on it in the “participants” section:

“The remaining 49 participants belonged to the high autistic-traits group had an average AQ of 26.61 (SD = 5.21). In other studies, subclinical groups of parents and siblings of individuals with ASD^{53,54} and specialists in mathematics⁵² were characterized by average AQs of 25.3 (SD = 10), 22.7 (SD = 5.5) and 24.5 (SD = 5.7), respectively. “

- Line 662 – it is unclear what is meant by one participant reporting mild depression in this context. How was this screened?

Every participant underwent a clinical interview. One participant reported that they experienced mild sadness. In order to be sure that this person didn't suffer from a depression, we used the Beck Depression Inventory (BDI) and the results indicated that this person did not have depression. We describe this process in the revised Table 1:

"1 In the high autistic-traits group, one participant reported sadness. In order to eliminate the possibility of depression, we asked this person to complete the Beck Depression Inventory (BDI, version 1978; Beck et al., 1988). This person scored 7 out of 63 points, which indicates no depression. "

- Line 773 – It would help here to clarify which tests were hypothesis driven and tested with one-tailed tests. I think it would be more standard to use two-tailed tests in this context.

We specified in the "Behavioral data analyses and statistics" section which tests had a directed hypothesis, based on our hypothesis (see last paragraph of introduction). (As we had specific hypothesis about the behavioral results we believe that the tests should be tested accordingly one-tailed or two-tailed):

"We calculated post-hoc tests one-tailed if we had a directed hypothesis and two-tailed if we assumed no differences between the autistic-traits groups: According to our hypothesis, we expected no differences between the autistic groups in the low load condition (i.e., two tailed testing), but a decrease in performance in the high load condition in the high autistic-traits group in comparison to the low autistic traits group (one-tailed). We did not expect a difference between the autistic-traits groups in the verbal task (two-tailed), but expected a decrease in performance in the visual and social task condition in the high autistic-traits group in comparison to the low autistic traits group (one-tailed)."

- Some other information in the Methods seems extraneous and could be streamlined to focus on the key scientific details.
For example, what defines a comfortable chair? How was it determined that it was comfortable?

Thank you, we streamlined the focus of the methods section and deleted unnecessary information.

Other (minor):

- There are some general language errors throughout the manuscript that should be fixed before publication. Some of these, e.g., the use of 'coherent' instead of 'consistent' alter the meaning of sentences and thus could cause confusion.

Thank you, we read-through the manuscript and improved the language.

Reviewer #2 (Remarks to the Author):

Dear Researchers

I deeply appreciate the opportunity to participate in the review of this research. I consider it to be an outstanding study that examines a broad sample of subjects using electroencephalography techniques. This approach is not only innovative but also explores traits related to the community, addressing potential conflicts linked to the field of mental health.

The work integrates behavioral aspects associated with cognitive functioning and intertwines them with actions and daily life behaviors, such as mentalization. Furthermore, it constitutes a compelling contribution to the field of neuroscience. However, certain aspects could be improved, which I have detailed in an attached file.

In the attached file, different questions have been answered and formulated to organize the information and establish a cohesive narrative that strengthens the presentation of this research. I hope these suggestions are taken into consideration with the aim of enhancing understanding among various readers and positioning this study as an innovative and relevant contribution to different communities.

Sincerely,

Thank you for the positive and constructive feedback. We have improved the organization, the narrative and the presentation of our study according to your suggestions and believe that we could strengthen the understanding and impact of our work.

Peer Review Feedback and Recommendations

1. General Aspects of the Document:

o **Study Summary:** The study investigates how slow rhythmic brain activity in the dorsomedial prefrontal cortex (DMPFC) regulates distributed networks associated with working memory and social cognition in individuals with varying levels of autistic traits. Brain activity was recorded from 100 volunteers, divided into two groups based on their autistic traits: low ($AQ \leq 20$) and high ($AQ > 20$). The results reveal that individuals with low autistic traits effectively utilize slow rhythmic activity to fine-tune brain communication, whereas those with high autistic traits face difficulties in this regulation, leading to deficits in social cognition and working memory. The study suggests that brain oscillations play a unifying role in cognitive coordination and provide explanations for challenges faced by individuals with high autistic traits in these areas.

o **Title:** The current title is descriptive, but a more concise version with a focus on key terms is recommended to enhance searchability. A suggestion:

"Dysregulated Neural Communication: A Common Marker of Social Cognition and Working Memory Deficits in Individuals with Elevated Autistic Traits."

Thank you for your suggestion, we have changed our title to:

"Dysregulated Control of Neural Communication: A Common Marker of Social Cognition and Visual Working Memory Deficits in Individuals with Elevated Autistic Traits"

o **Abstract:** The abstract does not explicitly outline the objectives, methods, results, and conclusions. To improve, structure the abstract to highlight the study's central idea and motivate the reader. Specifically, the phrasing: "We recorded electrical brain activity of volunteers spanning a wide range of autistic personality traits..." may introduce conceptual inaccuracies. Clarify whether the study addresses autistic spectrum conditions (Axis I in DSM-V) or personality-related traits (Axis II).

Thank you for this comment.

We clarified in the abstract that we did not record individuals with a certain Autism Spectrum Diagnosis (neither Axis I nor Axis II in the DSM-IV). We recorded subclinical individuals who differed in their extent of autistic personality traits:

"We recorded electrical brain activity from volunteers who differed in the extent of (subclinical) autistic personality traits."

Moreover, we re-organized the abstract as follows:

A clear objective in the beginning:

"It has long been thought that coordination of briefly maintained information (working memory) and social cognition (mentalizing) rely on different brain mechanisms. However, here we aimed to show (1) that visual working memory and social cognition share the same neural communication mechanism (i.e., interregional phase-amplitude coupling) and (2) that this mechanism is behaviorally relevant."

Then the Methods:

"We recorded electrical brain activity from volunteers who differed in the extent of (subclinical) autistic personality traits."

Followed by the results organized by objective:

“First, we showed that slow rhythmical brain activity in the dorsomedial prefrontal cortex controls distributed networks associated with working memory as well as mentalizing during cognitively demanding visual and social tasks. Depending on the effort necessary for cognitive operations, the phase of slow frontal oscillations is used to precisely tune communication with posterior brain areas. Individuals with low autistic personality traits use slow rhythmical brain activity in the dorsomedial prefrontal cortex as a general mechanism for tuning communication within this distributed network.

Second, individuals with high autistic personality traits struggle in finetuning this mechanism, which is associated with difficulties in efficiently adapting brain activity to the difficulty level of a visual working memory task; and they demonstrate problems with efficiently synchronizing the relevant cortical network in a social cognition task. “

And in the end the conclusion:

“While these findings suggest a unified function of brain oscillations in cognitive coordination between social and visual tasks, they also explain why individuals with high autistic personality traits can have difficulties with mentalizing and demanding cognitive processing.”

o **Structure:** The logical format should be more explicit. The introduction lacks a clear start, beginning with a metaphor as an example, which introduces some disorganization. Reorganize the introduction to define key concepts and provide relevant context. Highlight methodologies and their relevance to the research question.

We decided to stay with a metaphor in the beginning because we truly believe that readers can relate to the subject better with the metaphor and it attracts a broader readership.

However, we agree with your suggestions to improve the structure of our introduction and manuscript:

After the metaphor, we first explain the key concepts more comprehensively, then the gap in literature/problem and recent EEG studies to address it. Then, we explain the link to Autism Spectrum Disorder and end the introduction with the research question and hypotheses.

Moreover, we introduce “phase-amplitude coupling” and explain its relevance.

As we have re-structured the manuscript in a way that the methods are located before the result section in the new version, we could eliminate the other methodologies from the introduction, making it clearer.

o **Language and Style:** Adopt a more formal academic writing style to match the expectations of the target audience.

Thank you for this comment, we improved our writing style.

o **Originality:** The study is highly relevant, providing plausible explanations for a significant issue linked to autistic traits using non-invasive techniques such as EEG.

2. Introduction:

o **Clarity of the Problem:** The research problem could be articulated more clearly. Define the main concepts, such as working memory, to better guide the reader.

We improved the structure of the introduction and defined the main concepts and research problem more comprehensively:

“Difficulties with social cognition (i.e., cognitive processes needed to store social information and to apply it to oneself and others) can not only be challenging in daily life but they are also related to various psychiatric disorders ¹.”

“Attributing mental traits to another person, such as what kind of presents someone might like or dislike, strongly relies on activity within the so-called mentalizing network in the brain ^{2,3 4–6}. It includes a group of cortical regions strongly associated with social cognition in the dorsal parietal cortex, temporal cortex, cingulate cortex, and dorsal and medial prefrontal cortex ^{2–6}. However, usually, high-level social cognitive functioning also requires flexible transient storage and utilization of large quantities of information – processing known as working memory ⁷. Accordingly, in situations as described in the example above, one would also expect an involvement of the working memory system in addition to the mentalizing network. It is, however, unclear how far the mentalizing and the working memory networks can be overlappingly active in the brain. Whereas there is research suggesting that they show an antagonistic relationship ^{8,9}, Meyer and colleagues proposed that truly effortful social cognition relies on simultaneous activity of both fronto-parietal mentalizing and working memory networks ^{10–12}. ”

o **Context:** Enhance the discussion of the study’s relevance and the knowledge gaps it addresses.

We summarize the gap and importance right before the research questions:

“However, it is exactly this optimal allocation of neural resources depending on task-difficulty (in visual working memory processes) that is reflected by fine-tuned FM-theta phase to posterior gamma amplitude coupling, as discussed earlier ²³. With FM-theta being generated in the DMPFC that supposedly represents a hub region for coordinating mentalizing and working memory networks. There is also reason to believe that the phase-amplitude coupling mechanism responsible for coordinating communication between brain regions in visual working memory tasks can be extended to social cognition. This will shed light on the contradictory ideas how working memory and mentalizing are associated with each other relationship ^{8,9,11,62,63}. Moreover, if a common neural communication pathway is the underlying

cause for challenges in social cognition and visual working memory in ASD, this should also be reflected by FM-theta phase to posterior gamma amplitude coupling being associated with the extend of autistic traits in individuals.”

o **Objectives:** Explicitly state the study’s objectives and research questions, explaining how the chosen methodology addresses them. Refine the hypothesis for better organization and clarity.

In the end of our introduction, we have eliminated all unnecessary information and focused on the research question and hypotheses:

We now clearly state what research questions we want to answer, and the last paragraph clearly stated the hypotheses and relates them to each research question. The hypotheses refer to the most central methods used (phase amplitude coupling). For better clarity, the other methods are in the method section.

*“This raises two research questions: (i) if effortful social cognition requires cooperation between a fronto-parietal mentalizing network and a fronto-parietal working memory network, would those networks be coordinated by the above described FM-theta phase mechanism? And if so, (ii) could this **phase-amplitude coupling be behaviorally** relevant and explain why some people have difficulties in social cognition as well as **visual** working memory?”*

*To address research question (i), we expected that in a social as well as in a visual working memory task, fronto-parietal networks would be coordinated by FM-theta phase **from the DMPFC (in contrast to a verbal working memory task)**. We hypothesised that the mechanism of posterior brain activity nesting into different FM-theta phase segments depending on cognitive effort (see ²³) should be observable for the mentalizing network in the same way as for the working memory system in individuals with low autistic personality traits.*

Regarding research question (ii), we expected participants with high autistic traits to struggle with the social working memory task based on a failure of efficient nesting of posterior high frequency brain activity into the excitatory FM-theta phase. We also hypothesised that participants with high autistic traits would be more rigid in their dynamic control of fronto-parietal networks in the visual working memory task and would show deficits especially in the highly demanding level of the task.

*Data in line with these hypotheses would (i) provide evidence for a **common** mechanism of controlling **visual** working memory as well as mentalizing networks by the means of FM-theta oscillations. And (ii) they should explain why **social cognition as well as visual working memory processes** are aberrant in individuals with high autistic traits.”*

3. Methods:

o **Design:** Improve Figure 1 to better organize the steps and number of participants recruited. Include a comprehensive diagram illustrating the screening process and participant flow.

We fully revised Table 1 to include the screening steps and all of the participants information.

o **Procedures:** Describe the methods in more detail to ensure replicability.

We added a new figure, Figure 2, to enhance the clarity of what procedure was done at which stage and what was its main characteristics and outcomes. This detailed description will better ensure replicability.

o **Sample:** Present a table with descriptive analyses of the sample for each instrument used. First, explain the recruitment process and informed consent, followed by the applied instruments.

We present now the screening procedure and the descriptive analysis for each instrument in Table 1.

Moreover, we have re-structured our chapter “screening”: First, we explain the recruitment process and informed consent, followed by the applied instruments.

“Screening

Participants were mainly recruited within the LMU Munich and Technical University of Munich student communities and were compensated with course credit or 10 € per hour. All participants gave written informed consent prior to participation in the study, which was approved by the LMU Munich Faculty 11 ethics committee.

In our screening, first we used.....”

o **Analysis:** The analysis is robust but could be better structured around the primary outcomes.

We decided to structure the method section based on the procedure and not around the primary outcomes. We believe that this way, it is easier to replicate the methods, as it reads like a recipe you can follow step-by-step.

4. Results:

o **Clarity:** Organize the results more systematically to align with the objectives and research questions.

Thank you for this comment. As we have re-structured our manuscript that the methods are presented before the results, we could delete methodological information out of the results section, making it clearer. Additionally, we added the chapter “control analysis” and put all the analysis in there, leaving the primary outcomes more streamlined and cleaner.

We re-organized our result section in the primary outcomes and indicated which research question was addressed by which chapter:

“Low autistic-traits participants show expected phase-amplitude coupling (Research question 1)”

“High autistic-traits participants show a deviant brain communication pattern (Research question 2)”

“Circular-linear correlations between phase-amplitude coupling and personality traits (Research question 2)”

“Behavioral results (research question 2)”

Additionally, we added Figure 2 in order to make it clear, which analyses addressed which research questions.

o **Tables and Figures:** The images are excellent, but the legends should be clearer and more accessible for readers. Improve legends of Figure 3

We added a better legend referring to the theta trough at 180 degrees and we also added standard errors to the data points.

o **Relevance:** The results address the research questions effectively.

5. Discussion:

o **Interpretation:** Interpret the results more thoroughly within the theoretical framework.

o **Comparison:** Expand comparisons with previous studies and existing literature.

o **Implications:** Discuss the practical and theoretical implications of the findings in greater depth.

We have reviewed and changed the discussion section accordingly. Unfortunately, as our manuscript is already too long, we cannot add too many more studies.

o **Limitations:** Clearly acknowledge the study's limitations.

Thank you, we added a chapter in the end of the discussion section stating the limitations:

“Limitations

Our study was conducted using EEG. Although EEG has a very good temporal resolution, its spatial resolution is rather poor, and the signals are only recorded at the surface of the head. We used 3D source reconstruction to estimate the signals at specific regions of interest, however, this transformation is dependent on the implemented models. Intracranial recordings could better capture deep or spatially overlapping sources and magnetoencephalography (MEG) could better detect signals originating from the sulci. Accordingly, such alternative recording methods might have been able to fully explain why

the phase-amplitude mechanism was so specific to the DMPFC and not shown at the left DLPFC for the verbal task.

Despite this limitation, we were able to replicate - in a hypothesis-driven study with a prior defined ROIs and frequency bands - previous phase-amplitude coupling findings in the visual domain²³. Our control analyses for FM-theta and posterior gamma amplitude confirm that differences between the autistic-traits groups really do rely on the phase-amplitude coupling mechanism and cannot be explained by evoked activity and mere differences in amplitude which could lead to differences in signal-to-noise ratio and accuracy of phase estimates⁹⁷.

Another limitation of the present study is that the obtained circular correlations were not powerful enough to withstand multiple-testing corrections, despite our sample size of 100 participants. However, we could replicate memory-load dependent modulation in performance in all the working memory tasks^{10,88} And the reported findings were based on two groups of participants, who were carefully screened and matched in sociodemographics, medical history and intelligence in order to mainly differ in the extent of autistic traits.

Future studies are necessary to pin down the causal relationship between the reported phase-amplitude coupling mechanism and deviations in social cognition and working memory processes. Inhibiting the DMPFC with brain stimulation could also reveal whether the here reported phase-amplitude mechanism is essential for dynamic allocation of cognitive resources; and the spatial specificity of the effects (i.e. whether they really can only be obtained for DMPFC) could potentially be addressed using a neurostimulation approach.”

6. Conclusions:

o **Coherence:** Ensure the conclusions align with the objectives and findings.

We have re-structured the first paragraph to relate to the (i) and (ii) research objective:

“This study showed that (i) effortful social cognition is coordinated by FM-theta phase to posterior gamma amplitude coupling, similar to visual working memory tasks. In both tasks, DMPFC theta phase coordinated neural communication with the working memory and mentalizing system. Moreover, this study included participants with a wide range of autistic personality traits – from no signs to a high degree of autistic personality traits – who were comparable in age, gender, education and intelligence. Strikingly, differences in the extent of autistic personality traits - while controlling for other confounding variables - were enough to manifest themselves in different patterns of efficient coupling and decoupling of fronto-parietal networks.

Thus, (ii) the here identified phase-amplitude coupling mechanism was sensitive enough to reveal differences between subclinical individuals depending on the extent of their autistic personality traits and showed to be behaviourally relevant for difficulties in effortful social cognition as well as high load condition. “

o **Impact:** Highlight the study’s significance for the field.

In the second paragraph, we highlight the importance of the study:

“Moreover, this phase-amplitude coupling mechanism can be more or less fine-tuned, which can explain the different severity of deficits. Thus, the described phase-amplitude

mechanism *could* provide a unifying explanation for severe difficulties in social cognition and less severe difficulties in visual working memory in high load conditions for individuals on the autism spectrum. It might not only be able to explain the variety of symptoms within one disorder but also why these difficulties in cognition and social functioning often are shown in *a variety of* different mental disorders. We might be able to generalize our findings to other psychiatric and neurological conditions - such as schizophrenia - to *better* understand problems in broad cognitive control and social cognition ¹⁰⁴. *Recognizing this shared neural communication mechanism as reported in the current study could be a critical step toward understanding the underlying cause of symptoms and developing cross-diagnostic interventions to improve therapies for a range of mental health conditions.*”

o **Suggestions:** Propose clear directions for future research and potential applications of the findings.

In the third paragraph, we address the future directions:

“Future studies *should therefore* translate this phase-amplitude coupling mechanism into clinical applications. *As the described phase-amplitude coupling is a measure of neural communication, it might reflect the complexity of network processes better than amplitude changes at single electrodes. Therefore, it has the potential for more innovative and improved applications in the following fields:* First, by taking the developmental component into account ⁵⁸, this phase-amplitude coupling mechanism might serve as an early predictor for aberrant development. Second, it could operate as neural signature to evaluate therapeutic approaches such as neurofeedback training, cognitive training or pharmaceutical approaches in pre-post tests. Third, innovative neurostimulation interventions could be developed based on our results to improve working memory and social cognition ^{28,105}. “

7. References:

o **Recency:** Ensure the references are current and relevant.

The references are current and relevant.

o **Citations:** Verify that all citations are properly referenced in the text.

All citations are properly referenced in the text.

o **Format:** Follow the journal’s bibliographic style consistently. It appears the current format does not align with APA 7th edition standards.

Communications Psychology uses standard Nature referencing style, which we consistently follow.

8. Recommendations and General Evaluation:

o **Strengths:** The study’s sample and methodology are commendable, as is the application of EEG techniques to address complex traits associated with mental health.

o **Areas for Improvement:** While the methodology is strong, other sections (e.g., introduction, results) require better organization to align with specific objectives.

We re-organized the introduction and the result section according to your suggestions above and believe our manuscript has substantially improved.

o **Constructive Comments:** Provide clear responses to the questions outlined in this review. Organize results by specific objectives.

We organized the result section by objectives and additionally added Figure 2 , which outlines the analyses and relates the primary outcomes to the objectives.

o **Final Recommendation:** Accept with major revisions

Reviewer #4 (Remarks to the Author):

This paper examines fronto-parietal phase-amplitude coupling (PAC) during three types of working memory tasks (visual, verbal, social) and how it depends on autistic traits. The authors focus on frontal theta phase and its modulation of parietal gamma power, a choice that is well motivated and described in detail. They report some differences in both the magnitude and phase of PAC between individuals divided into groups with low and high autistic scores, respectively.

The rationale behind the study is described clearly and corresponding hypotheses are tested with advanced statistical methods. I also appreciated the large sample size and thorough analysis of the data. However, I do have some concerns about the interpretation of the results and some methodological questions.

As stated in the title, the authors claim to show a “common signature for [...] social cognition and working memory”, and how it is changed in the presence of strong autistic traits. Apart from the title, this claim appears throughout the manuscript, e.g., “FM-theta phase does not only seem to control the fronto-parietal working memory system but also the mentalizing network”, “shared neural bases between the visual and social tasks”, etc. This claim puzzles me as the three tasks are all variants of a working memory task. How do the authors’ results demonstrate a functional role of fronto-parietal PAC that goes beyond working memory? How can the “mentalizing network” be identified in the data? I think the manuscript would benefit from a more precise description of the cognitive functions the three tasks are designed to test, and what the obtained results mean in this respect.

Thank you for your insightful comments that made us aware of issues requiring critical clarification.

The dorsomedial prefrontal cortex (DMPFC) has been described as a hub region for social cognition as well as a key region for (non-social) working memory processing. This makes the DMPFC a prime candidate for linking the mentalizing network and the working memory system. That is why we recorded FM-theta from the DMPFC. Moreover, our results showed that the DMPFC controls posterior gamma amplitude from regions usually associated with the working memory system as well as regions associated with the mentalizing system. And this is true for the visual and social tasks but not for the verbal task.

If the phase-amplitude coupling were only a function of working memory processes, then the phase-amplitude coupling mechanism should have been the same in all three working memory tasks (also in the verbal one) and FM-theta should have controlled only posterior regions associated with the working memory system (but not posterior regions associated with the mentalizing system).

We clarified this in the introduction and in the discussion section. We also added a more precise description of the tasks in the introduction section.

Please see also the comments to your next point below:

It also remains unclear to me how the authors interpret the null effect in the verbal task. If fronto-parietal PAC underlies social and visual working memory, but not verbal working memory, what does that mean for its role in the brain? I appreciate the explanation that the

authors propose in the Discussion, but it doesn't seem to be confirmed in the additional analysis they describe. I'm therefore missing a conclusion of whether PAC is a general mechanism (present for more than one task) or a specific one (absent in one task). Irrespective of this conclusion, I would find it helpful for the reader to see results for PAC in the verbal memory task as well.

Thank you for this comment! It made us realize that – in an attempt to shorten the manuscript – we have been not clear enough about our hypotheses and conclusions of the verbal task.

First, we added a more precise description of the verbal task in the introduction:

“Controlling a distributed verbal working memory network³¹ seems to be slightly different than coordinating mentalizing and visual working memory networks. There, the left dorsolateral prefrontal cortex (DLPFC) seems more relevant than the DMPFC^{10,32–35}. In brain stimulation studies, the inhibition of the left DLPFC led to a performance decrease in a broad variety of tasks, suggesting a more general working memory function^{34–36}. In contrast, an artificial lesion over the DMPFC was found to impact performance in visual working memory as well as mentalizing tasks (e.g.,^{36,37}).

Consequently, while verbal working memory seems to recruit a distributed cortical network with utmost importance of the DLPFC, the visual working memory and mentalizing tasks might additionally recruit networks responsible for detecting, attending and processing visual stimuli, which also act as social cues (i.e., detecting gaze directions, reading facial expressions, reorienting visual attention)^{39,40}. And those regions are under strong influence of the DMPFC. “

We moved the analyses of the left DLPFC and the verbal task to the result section:

“Left DLPFC theta phase to posterior gamma amplitude coupling

As a range of studies suggest that verbal working memory might rely more strongly on the left dorsolateral prefrontal cortex (DLPFC)^{10,32–35} than on the DMPFC, we analyzed the data with exactly the same procedure as reported above but used the left DLPFC instead of the DMPFC for theta phase extraction (see Table S3). Regression models with theta phase segments from the left DLPFC, load and autistic-traits group as interaction terms suggest significant effects for 50-Hz posterior gamma amplitude in the verbal (AICc = 40.40) and visual (AICc = 28.07) tasks but not in the social task (AICc = -14.79 to -1.75, n.s.). However, the anticipated phase-amplitude coupling (as obtained with DMPFC theta activity for individuals with low autistic traits), could not be systematically found for the left DLPFC (see Figure S5 and S6).

This suggests that the effects from the main analyses as described above are specific for theta phase obtained from DMPFC and selective for only the visual and the social tasks.”

The phase-amplitude coupling figure for the verbal task is in the Supplementary material, Figure S5 for the left DLPFC. DMPFC theta phase to posterior gamma amplitude coupling cannot be illustrated as the regression analyses did not yield significance.

We revised the manuscript carefully to state clearly if the conclusions were true for all tasks or not and we specified our conclusions in the discussion section:

“(i) Low autistic-traits participants show unified visual working memory and mentalizing networks

Posterior working memory and mentalizing regions were co-activated and controlled by the identical prefrontal oscillatory mechanism during social as well as visual working memory tasks in individuals with low autistic personality traits: “

.....

“In contrast, the verbal task did not elicit the same phase-amplitude coupling as seen in the other two tasks. This was not only true for the DMPFC (as hypothesized) but also for the left DLPFC (in contrast to our hypotheses).

One explanation could be, that even if the phase-amplitude mechanism extended to the left DLPFC the applied methods might not have picked up the relevant signal for identifying such: Due to the anatomical structure of the cortex, more complex cortical folding in the lateral prefrontal regions, might lead to dipole cancellation and increased variability across individuals and, thus, problems in consistently detecting phase-amplitude coupling with surface EEG in the DLPFC. A study by Canolty and colleagues²¹ supports this idea. Canolty et al. used electro-corticographic (ECoG) recordings from the DLPFC while participants performed a range of cognitive tasks. ECoG is directly recorded from the cortical surface, which leads to a better signal-to-noise ratio and less dipole cancellation. And indeed, the authors report consistent phase-amplitude coupling between DLPFC-theta and gamma activity. Another important finding that Canolty and co-workers reported is that similar cognitive tasks lead to similar phase-amplitude coupling patterns. In respect to the present study this suggests that from a neural perspective the social cognition task and the visual working memory task were processed highly similar, whereas the verbal working memory task, apparently is solved entirely different by the brain.”

The authors' analysis relies heavily on source-localised EEG data without the use of individual anatomical scans to improve the spatial accuracy (if I understand correctly). Is there any way the authors can estimate how well the DMPFC was localised? I wonder whether this could be addressed by showing that the observed theta-gamma PAC is specific for the ROIs selected and does not occur between other regions.

Yes, we showed that this phase-amplitude coupling mechanism was specific for the DMPFC and not found for the left DLPFC for example. We describe the results in a new chapter after Control analyses in the result section:

“Left DLPFC theta phase to posterior gamma amplitude coupling

As a range of studies suggest that verbal working memory might rely more strongly on the left dorsolateral prefrontal cortex (IDLDFC)^{10,32–35} than on the DMPFC, we analyzed the data with exactly the same procedure as reported above but used the left DLPFC instead of the DMPFC for theta phase extraction (see Table S3). Regression models with theta phase segments from the left DLPFC, load and autistic-traits group as interaction terms suggest significant effects for 50-Hz posterior gamma amplitude in the verbal (AICc = 40.40) and visual (AICc = 28.07) tasks but not in the social task (AICc = -14.79 to -1.75, n.s.). However, the anticipated phase-amplitude coupling (as obtained with DMPFC theta activity for individuals with low autistic traits), could not be systematically found for the left DLPFC (see Figure S5 and S6).

This suggests that the effects from the main analyses as described above are specific for theta phase obtained from DMPFC and selective for only the visual and the social tasks..”

Additionally, we acknowledged the drawback of EEG in the limitation section:

“Our study was conducted using EEG. Although EEG has a very good temporal resolution, its spatial resolution is rather poor, and the signals are only recorded at the surface of the head. We used 3D source reconstruction to estimate the signals at specific regions of interest, however, this transformation is dependent on the implemented models. Intracranial recordings could better capture deep or spatially overlapping sources and magnetoencephalography (MEG) could better detect signals originating from the sulci. Accordingly, such alternative recording methods might have been able to fully explain why the phase-amplitude mechanism was so specific to the DMPFC and not shown at the left DLPFC for the verbal task.

Despite this limitation, we were able to replicate - in a hypothesis-driven study with a prior defined ROIs and frequency bands - previous phase-amplitude coupling findings in the visual domain²³. Our control analyses for FM-theta and posterior gamma amplitude confirm that differences between the autistic-traits groups really do rely on the phase-amplitude coupling mechanism and cannot be explained by evoked activity and mere differences in amplitude which could lead to differences in signal-to-noise ratio and accuracy of phase estimates⁹⁷.”

Results:

“regression models for the different working memory tasks with FM-theta phase segments [...] as predictor variables”

Could the authors please provide more details on the regression model used? It is unclear to me how this model includes theta phase. Is it used as a circular predictor in a linear regression model?

The purpose of the regression analyses was to determine whether there is a difference between the theta-phase sorted gamma amplitudes (criterion) in relation to the predictors theta phase, load, a prior defined posterior ROIs and autistic groups. If the predictor theta phase is significant it means that the gamma amplitude is different over different theta phase bins (without an assumption how this difference is).

To make that clearer, we added Figure 2 and reorganized this paragraph as follows:

“As described in the method section, we sorted posterior gamma amplitude as a function of FM-theta phase. The sorted posterior gamma amplitude values were used as dependent variable in regression models for the different working memory tasks with FM-theta phase segments, load, posterior ROIs and autistic-traits groups as predictor variables. If gamma amplitude is equally distributed across FM-theta phases, this would indicate no association between posterior gamma amplitude and FM-theta phase. If, however, sorted gamma amplitude systematically varies across different FM-theta phases, it will suggest interaction between FM-theta phase segments and posterior gamma (see²³ for details and Figure 2). Thus, only significant interactions involving the predictor FM-theta were indicative of posterior gamma amplitude being modulated by theta phase segments and thus were taken into account.”

Additionally, we realized that it is not clear enough to only mention in this first paragraph above, that only significant interactions involving the predictor FM-theta were taken into

account. Thus, we explicitly named the predictor FM-theta in every regression model in the following description for clarity:

“Regression models on FM-theta sorted gamma amplitude indicated significant interactions between FM-theta phase segments, load and autistic-traits group for the social task (at 70 Hz: second-order Akaike information criterion value (AICc) = 34.51; at 60 Hz: AICc = 20.84; note, AICc values higher than 10 are considered as significant, see methods section for more information) and the visual task (70 Hz: AICc = 17.66). These effects generalized across posterior ROIs. There was no effect for FM-theta phase segments x posterior ROIs (AICc = -109.90 to 2.99, n.s.), FM-theta phase segments x posterior ROIs x load (AICc = -249.16 to -131.95, n.s.), FM-theta phase segments x posterior ROIs x autistic-traits group (AICc = -232.44 to -126.13, n.s.) or FM-theta phase segments x posterior ROIs x load x autistic-traits group (AICc = -500.64 to -318.89, n.s.) in any of the gamma frequency bands. This result indicates that gamma amplitude was not statistically significant different modulated by FM-theta phase at the different posterior ROIs (i.e., from the mentalizing network as well as those from the working memory system). Since the model indicated that the factor “posterior ROIs” did not contribute meaningful information, retaining this factor in the model was not justified. This is why for any further analyses, sorted gamma amplitudes were averaged across all posterior ROIs.”

After this difference was statistically confirmed, we went to the second step: In the second step, we investigated whether the differences followed a circular cosine model or not. This was done by the Cosine Model Fitting.

In order to make these steps clearer, we have added Figure 2.

“[...] was similarly modulated by FM-theta phase”

I might have misunderstood something but I'm missing the effect of theta phase in the description of the statistical results (i.e. theta phase predicting gamma power).

Thank you for raising this issue. The effect of theta phase was statistically tested with the regression models:

“The sorted posterior gamma amplitude values were used as dependent variable in regression models for the different working memory tasks with FM-theta phase segments, load, posterior ROIs and autistic-traits groups as predictor variables. If gamma amplitude is equally distributed across FM-theta phases, this would indicate no association between posterior gamma amplitude and FM-theta phase. If, however, sorted gamma amplitude systematically varies across different FM-theta phases, it will suggest interaction between FM-theta phase segments and posterior gamma (see ²³ for details and Figure 2). Thus, only significant interactions involving the predictor FM-theta were indicative of posterior gamma amplitude being modulated by theta phase segments and thus were taken into account.”

As described above, we clarified in the chapter “DMPFC theta phase to posterior gamma amplitude coupling” in the result section that all described effects included a significant interaction with FM theta phase segments.

Fig. 3: Would the authors please include a measure of between-subject variance?
Thank you, we added the standard error in the Figure.

Currently data from the two subject groups with high and low autistic traits seem to be analysed separately, with the exception of the circular-linear correlation. Given the authors' conclusion of differences in PAC between groups, a direct contrast between the two (predictive power of theta phase in the regression model, statistical contrast of phase etc) seems important to draw this conclusion.

We draw this conclusion based on the regression models. The regression models indicate a significant interaction between the autistic-traits group and the load condition in addition to the theta phase. This is a direct statistical contrast between groups. Only based on this statistically significant result, we show the cosine model fits in Figure 3 separate for autistic-traits group and load conditions. The model fits have to be performed separate for each group and load condition in order to see the difference between them and to be able to compare them to our hypothetical model derived from Berger et. al., 2019.

We added Figure 2 to illustrate the steps better and tried to make that more specific in the results section:

“Thus, the results from the regression models indicate that a statistically significant interaction between autistic-traits groups and load conditions are predictive of gamma amplitude associated with certain theta phases. In order to further characterize the nature of the significant modulation of FM-theta phase-sorted gamma amplitude, we fitted our sorted gamma amplitude values to a theta cosine model: (1) This cosine model was compared to an intercept model to check if the modulation of gamma amplitude was significant, separately for respective load condition and autistic-traits group (AIC model comparison cosine vs. intercept). (2) We computed the mean absolute error (MAE) in order to determine how well our empirical data fit to a cosine model and report the confidence interval range. A good model fit indicates that posterior gamma amplitudes are periodically modulated by FM-theta phase. (3) The cosine model was compared to a null-shift reference cosine model with the trough at 180° to establish whether posterior gamma was precisely locked to the FM-theta trough or to any other FM-theta phase segment.”

[...] “these results should be considered with caution as the correlations were not corrected for multiple testing”. Could the authors please specify how many tests were made and/or include statistics for a version corrected for multiple comparisons?

For the three significant PAC results (social task (60 Hz and 70 Hz) and visual task 70 Hz), we calculated for both load conditions the correlation with all 5 subscales of the Autism Spectrum Quotient over all participants. That means $3 \times 2 \times 5$ tests = 30 tests).

Additionally, we calculated the same correlation for reaction time and accuracy. That means $3 \times 2 \times 2 = 12$ tests.

We added this in the Methods Section, circular-linear correlations:

“These correlations were only calculated for the tasks and frequency bands showing a significant phase-amplitude coupling (total of 42 correlations).“

PAC can also arise from stimulus-evoked responses and is influenced by their shape (Lozano-Soldevilla et al., 2016, *Frontiers*). Can the authors rule out such an effect in their data, e.g., spurious PAC produced by responses evoked by instruction offset that differ between conditions? Given that the interval analysed is 2.5 s long, this seems relatively unlikely, but it would be worth showing how these evoked responses look like for the different conditions.

Thank you for this comment. We performed control analyses and added them in the result section under “Control Analyses”:

(Please see also Figure S4 for more details in the supplementary material)

“Fourth, we tested whether evoked responses – although unlikely due to our rather lengthy analysis time segment of 2.5 s - could have had any impact on phase-amplitude coupling results⁹⁸. Therefore, we shifted theta phase values and gamma amplitude by one trial, so that theta phase from trial 1 was now coupled with gamma amplitude from the last trial and theta phase from trial 2 was coupled with gamma amplitude from the first trial. This way only stimulus evoked effects remained in the data without any induced effects surviving the re-alignment. We then calculated the same phase-amplitude coupling values, regression models, and cosine fitting models. The results indicated that - unlike in the original analyses - no significant interaction in the regression models between FM-theta phase segments, load and autistic-traits groups in the social task in the 60-Hz ($AICc = -15.15$, n.s.) and 70-Hz gamma amplitude ($AICc = -33.84$, n.s.) were found. In the visual task, a significant interaction was obtained in the 70-Hz gamma amplitude ($AICc = 22.92$). However, in contrast to the results from the original analysis, gamma amplitude was not significantly periodically modulated by FM-theta or showed the reversed coupling pattern than in the main analyses (see Figure S4). These results indicate that the findings from the main analyses cannot be explained by mere stimulus evoked effects.”

The authors describe two types of changes in PAC: A phase shift and a change in magnitude. I think the manuscript would benefit from a clearer distinction of the two effects in the description of the results, and a speculation of how the two might differ in their functional significance.

Thank you for this suggestion. We added Figure 2 to make this issue clearer and explained in the Figure Legend:

“(B) illustrates the performed analyses with the theta phase-sorted gamma amplitudes and which of the two research questions they addressed. The combination of regression analyses and fitting steps (1) and (2) of the cosine model fitting can be interpreted as a measure of coupling magnitude: If there is no difference in coupling magnitude between conditions, the regression will not yield significant results. However, a significant regression does not necessarily imply a difference in phase-amplitude coupling magnitude, as the cosine model fitting steps (1) and (2) need to verify whether the coupling truly follows the theta phase. Step (3) of the cosine model fitting determines the shift of the coupling.”

Title: I suggest to avoid the term “communication” as it is not shown in the data.

We changed the title to “*Dysregulated Control of Neural Communication: A Common Marker of Social Cognition and Visual Working Memory Deficits in Individuals with Elevated Autistic Traits*”.

We decided to leave the word “communication” in the title and to better explain what we mean with it throughout the manuscript:

We analyze interregional phase-amplitude coupling as a measure of coordination of neural communication between distributed cortical regions during demanding tasks (i.e., the DMPFC locks posterior gamma amplitude to certain phase segments in order to optimize communication between frontal and posterior regions in the working memory and mentalizing system).

“i.e. roughly 5 theta cycles / s x 2.5 s manipulation period x 30 trials = 375 data points per phase angle, resulting in over 13 000 data points per analyzed phase segment). This number of data points ensures a robust analysis even with a relatively low number of trials.”

I did not fully understand the purpose of this sentence. As the authors use 7-cycle wavelets, the 5 theta cycles do not seem treated independently from each other. It is also unclear what the 13000 data points mean, is it the number of EEG sample points analysed? I suggest to either provide more details or remove the two sentences altogether.

Thank you, as our manuscript is already too long and this detail is not as important, we decided to delete this information.

Dear Reviewers and dear Editor,

Thank you for reviewing our manuscript. We revised it accordingly and hope that it is now suitable for publication in *Communications Psychology*.

Please find our changes in red in the manuscript and our answers to your suggestions below in blue.

Dear Professor Friedrich,

Thank you for your patience during the peer-review process. Your manuscript titled "Dysregulated Control of Neural Communication: A Common Marker of Social Cognition and Visual Working Memory Deficits in Individuals with Elevated Autistic Traits" has now been seen by 3 reviewers, and I include their comments at the end of this message. They find your work of interest but raised some important points. We are interested in the possibility of publishing your study in *Communications Psychology*, but would like to consider your responses to these concerns and assess a revised manuscript before we make a final decision on publication.

We therefore invite you to revise and resubmit your manuscript, along with a point-by-point response to the reviewers. Please highlight all changes in the manuscript text file.

Editorially, we consider it crucial that Reviewer #1's remaining concerns regarding the definition of autism and the analyses are thoroughly addressed in the revised manuscript.

I am attaching an Editorial Requests Table that details critical reporting requirements for the revised manuscript. Please attend to each item and ensure your manuscript is fully compliant. If your revised manuscript is not aligned with these requests on major issues, such as those concerning statistics, it may be returned to you for further revisions without re-review.

Best regards,

Troby Lui

Troby Lui, PhD
Associate Editor
Communications Psychology

Thank you, Dr. Troby Lui, for your valuable comments. We have addressed all issues raised by the reviewers and by the Editorial Requests Table in our re-submission.

REVIEWER REPORTS:

Reviewer #1 (Remarks to the Author):

I appreciate the authors' responsiveness to reviewer feedback and revisions to the manuscript. The revised manuscript is substantively improved. However, I have several remaining concerns that should be addressed before the manuscript is suitable for publication.

- The hypothetical at the beginning is still confusing. I don't understand how it links into the key concepts in the paper. I'm particularly baffled by the leap from 'awkward work Christmas party' to psychiatric disorders. It's also not a culturally sensitive example.

We have adapted our hypothetical example in the beginning to make it easier understandable and culturally sensitive:

"Imagine you are in charge of assigning the best-suited welcome present to each of your friends at a party. For this, you will need to infer from your friends' personality traits which type of present they might like; you will have to hold this information in memory and mentally combine friend-present pairings. Depending on how well you do these complex cognitive operations, the presents will either be a success or disappointment.

These operations - known as social cognition - involve storing social information and applying it to oneself and others. While subtle miscalculations can happen to everyone and may cause everyday uncomfortable moments, more persistent or severe difficulties in social cognition can have significant consequences and can be observed in various psychiatric disorders¹. Thus, the investigation of the underlying neural mechanisms of social cognition is a crucial and clinically highly relevant research topic."

- The revised definition of autism remains inaccurate and misleading. This needs to be fixed before publication. Furthermore, reading through the revised manuscript, I wonder if the authors are solely interested in autistic traits, and not ASD? Perhaps it would make sense to reframe the manuscript throughout to focus on autistic traits only.

Our manuscript focuses on autistic traits. However, we think it is important to introduce the concept of autism spectrum in order to understand what we mean with autistic traits. We have changed our introduction to the autism spectrum as follows:

"Autism is a spectrum and includes a wide range of characteristics (Lord et al., 2028, Lancet). In some cases, these characteristics are clinically significant and lead to a formal diagnosis. But autistic traits can also be present in individuals who do not meet the full diagnostic criteria (i.e. sub-clinical range). As shown by a meta-analysis by Rødgaard et al. (2019), the differences between individuals with and without a diagnosis have decreased

over time, highlighting the importance of using subclinical trait measures to capture the whole spectrum. For this reason and also to control for possible confounding factors such as medication intake or co-occurring conditions, we recruited a sub-clinical sample.”

- The authors stated that they ‘adopted a combination of person-first and identity-first language in this paper to acknowledge and reflect the diverse preferences within the community.’ However, after reading the paper, I only found one instance of identity-first language, with the paper otherwise written entirely with person-first language. Focusing on autistic traits would resolve this issue too.

Thank you, we have changed “people with autism/ASD” to “autistic people” throughout the manuscript as suggested by you, reviewer 2 and the paper by Bottema-Beutel et al., 2021, Autism in Adulthood.

The term “individuals with a diagnosis of ASD” (in contrast to individuals without a diagnosis) is only used in relation to the screening process in the methods section in order to be in accordance with official classification terms (e.g., ICD).

- The rationale for dichotomizing participants provided by the authors in their response actually reads like more support for not dichotomizing participants. Based on this, I think it would make sense to use the AQ as a continuous measure, at least in supplementary analyses.

We recruited the participants specifically to fit either to group 1 or group 2 based on our *a priori* defined study design, hypotheses and literature-based *a priori* defined cut-off. That means

- (1) Our study design is based on AQ as a between-subject variable. Therefore, also our power analysis is based on this difference hypothesis. We believe that modifying this approach post hoc would not be methodologically appropriate and could be considered inconsistent with good scientific practice.
- (2) We forced the AQ variable to have two distinct distributions (one around the mean of group 1 and the other around the mean of group 2). That means, our AQ variable is not a continuous, random variable following ONE normal distribution, which would be appropriate to use in our regression model.
- (3) Our hypothesis is that there are differences between the two AQ groups. If we included the AQ variable as continuous measure in the regression analysis, this AQ variable could not explain any variance as it would be the same in all conditions (tasks and loads) as all conditions were performed by the same participants (conditions = within-subject factor).

For these reasons, we cannot include the AQ as a continuous measure in our main regression analyses.

- The application of one-tailed post hoc tests in this context is not conventional. While further clarification has been provided about hypotheses that were directional, these contrasts weren’t pre-registered and could inflate Type I error. Please at least report two-tailed results in the supplement.

Two-tailed results can be obtained by multiplying the p-value:

$p \text{ (two-tailed)} = p \text{ (one-tailed)} * 2 \text{ (if direction is expected)}$

- The authors report that all participants underwent a 'clinical interview,' but also state that the Beck Depression Inventory (BDI) was used to 'eliminate the possibility of depression' in one participant who 'reported sadness.' This is unclear for several reasons. First, sadness is a normal human emotion and does not indicate depression in the absence of other clinical symptoms. Second, the BDI is a self-report symptom inventory, not a diagnostic tool. If a diagnostic clinical interview was conducted, it is unclear why the BDI would be needed afterward to confirm the absence of depression. The rationale for using the BDI in this context should be clarified.

Thank you for this comment. Yes, we agree that sadness is a normal human emotion and that it doesn't mean that the person is depressed (which he/she wasn't). We used the BDI only as an add-on and an attempt to make the emotion "sadness" more objectively measurable. In order to avoid confusion and as our manuscript pushes the limits of the allowed length, we decided to delete this information (which is not relevant for the conclusion of the study) from the Figure.

- The revised text still does not explain why mathematics specialists are of particular relevance for comparison in this context or justify the AQ cut-offs. I suggest removing this entirely.

We removed this sentence entirely from the manuscript: *"In other studies, subclinical groups of parents and siblings of individuals with ASD^{70,71} and specialists in mathematics⁶⁶ were characterized by average AQs of 25.3 (SD = 10), 22.7 (SD = 5.5) and 24.5 (SD = 5.7), respectively."*

- The authors elaborated on Berger et al, which was helpful. However, they still using the term 'exactly replicating'. Is this an exact or conceptual replication?

We apologize for being not precise enough with this term and agree that the replication is rather conceptual than exact. We corrected our sentence from "exactly replicating" to *"conceptually reproducing"*.

- Was surrogate testing performed as part of the PAC analysis?

No, we did not perform surrogate testing but chose a modelling approach. Please see explanation below.

- The PAC calculation is non-conventional. Could a comparison with traditional methods be added to the supplement?

Thank you for this suggestion. The reason for our approach is that our signal is a circular signal. And in order to answer our questions, we need to test the modulation of the circular signal, i.e., we need to find out whether gamma amplitude is modulated by theta phase.

Traditional methods, such as post-hoc t-tests in order to find whether gamma amplitude is significantly higher in theta phase bin 1 than in theta phase bin 2 for example cannot answer this question. We need to know HOW the signal is modulated over the theta cycle in order to answer our question, not whether it is differently modulated over a theta cycle. And in order to know whether our theta phase modulates gamma in a meaningful way, we need to model the data and test whether it follows a cosine curve or not. We provide confidence intervals for the results as traditional methods.

Reviewer #2 (Remarks to the Author):

I believe the changes made in response to the previous review have been beneficial, leading to improved analyses and a clearer overall structure of the manuscript. The article presents a novel approach that will certainly be of interest to the scientific community, particularly for its contribution to understanding the neurocognitive mechanisms associated with autistic traits. The conclusions appear original and well supported.

However, I would like to raise a point concerning the terminology used to refer to the studied population. In the current field of autism research, there is a growing tendency to use the term “autistic people” instead of “people with autism”, in alignment with the preferences expressed by many individuals on the autism spectrum and their communities. While I understand that the study focuses on individuals with autistic traits rather than clinical diagnoses, I encourage the authors to explore this discussion further. If supported by recent literature, I suggest making a small but meaningful adjustment to the way the population is referred to in the manuscript, out of respect for the community being studied.

All other revisions have been adequately addressed. The methodology is now clearly structured, which enhances the reproducibility of the study. The systematization of procedures adds rigor and facilitates replication, which are essential qualities for research of this nature.

In my opinion, the manuscript is suitable for publication.

Thank you, we have changed “people with autism” to “autistic people” as suggested.

Reviewer #4 (Remarks to the Author):

Thank you for this thoroughly revised manuscript. My questions have been addressed.